# Incomplete inhibition of HIV infection results in more HIV infected lymph node cells by reducing cell death

Laurelle Jackson[1,2†], Jessica Hunter[1,2†], Sandile Cele[1], Isabella Markham Ferreira[1,2], Andrew C Young[1,3], Farina Karim[1], Rajhmun Madansein[4,5], Kaylesh J Dullabh[4], Chih-Yuan Chen[4], Noel J Buckels[4], Yashica Ganga[1], Khadija Khan[1], Mikael Boulle[1], Gila Lustig[1], Richard A Neher[6,7], Alex Sigal[1,2,8]*

[1]Africa Health Research Institute, Durban, South Africa; [2]School of Laboratory Medicine and Medical Sciences, University of KwaZulu-Natal, Durban, South Africa; [3]Department of Neurology, Massachusetts General Hospital and Harvard Medical School, Boston, United States; [4]Department of Cardiothoracic Surgery, University of KwaZulu-Natal, Durban, South Africa; [5]Centre for the AIDS Programme of Research in South Africa, Durban, South Africa; [6]Biozentrum, University of Basel, Basel, Switzerland; [7]SIB Swiss Institute of Bioinformatics, Basel, Switzerland; [8]Max Planck Institute for Infection Biology, Berlin, Germany

**Abstract** HIV has been reported to be cytotoxic in vitro and in lymph node infection models. Using a computational approach, we found that partial inhibition of transmissions of multiple virions per cell could lead to increased numbers of live infected cells. If the number of viral DNA copies remains above one after inhibition, then eliminating the surplus viral copies reduces cell death. Using a cell line, we observed increased numbers of live infected cells when infection was partially inhibited with the antiretroviral efavirenz or neutralizing antibody. We then used efavirenz at concentrations reported in lymph nodes to inhibit lymph node infection by partially resistant HIV mutants. We observed more live infected lymph node cells, but with fewer HIV DNA copies per cell, relative to no drug. Hence, counterintuitively, limited attenuation of HIV transmission per cell may increase live infected cell numbers in environments where the force of infection is high.
DOI: https://doi.org/10.7554/eLife.30134.001

*For correspondence:
alex.sigal@k-rith.org

†These authors contributed equally to this work

## Introduction

HIV infection is known to result in extensive T cell depletion in lymph node environments (*Sanchez et al., 2015*), where infection is most robust (*Brenchley et al., 2004*; *Doitsh et al., 2010*; *Doitsh et al., 2014*; *Finkel et al., 1995*; *Galloway et al., 2015*; *Mattapallil et al., 2005*). Depletion of HIV infectable target cells, in addition to onset of immune control, is thought to account for the decreased replication ratio of HIV from an initial peak in early infection (*Bonhoeffer et al., 1997*; *Nowak and May, 2000*; *Perelson, 2002*; *Phillips, 1996*; *Quiñones-Mateu and Arts, 2006*; *Ribeiro et al., 2010*; *Wodarz and Levy, 2007*). This is consistent with observations that individuals are most infectious in the initial, acute stage of infection, where the target cell population is relatively intact and produces high viral loads (*Hollingsworth et al., 2008*; *Wawer et al., 2005*).

T-cell death occurs by several mechanisms, which are either directly or indirectly mediated by HIV infection. Accumulation of incompletely reverse transcribed HIV transcripts is sensed by interferon-γ–inducible protein 16 (*Monroe et al., 2014*) and leads to pyroptotic death of incompletely infected

**eLife digest** The HIVvirus infects cells of the immune system. Once inside, it hijacks the cellular molecular machineries to make more copies of itself, which are then transmitted to new host cells. HIV eventually kills most cells it infects, either in the steps leading to the infection of the cell, or after the cell is already producing virus. HIV can spread between cells in two ways, known as cell-free or cell-to-cell. In the first, individual viruses are released from infected cells and move randomly through the body in the hope of finding new cells to infect. In the second, infected cells interact directly with uninfected cells. The second method is often much more successful at infecting new cells since they are exposed to multiple virus particles.

HIV infections can be controlled by using combinations of antiretroviral drugs, such as efavirenz, to prevent the virus from making more of itself. With a high enough dose, the drugs can in theory completely stop HIV infections, unless the virus becomes resistant to treatment. However, some patients continue to use these drugs even after the virus they are infected with develops resistance. It is not clear what effect taking ineffective, or partially effective, drugs has on how HIV progresses.

Using efavirenz, Jackson, Hunter et al. partially limited the spread of HIV between human cells grown in the laboratory. The experiments mirrored the situation where a partially resistant HIV strain spreads through the body. The results show that the success of cell-free infection is reduced as drug dose increases. Yet paradoxically, in cell-to-cell infection, the presence of drug caused more cells to become infected. This can be explained by the fact that, in cell-to-cell spread, each cell is exposed to multiple copies of the virus. The drug dose reduced the number of viral copies per cell without stopping the virus from infecting completely. The reduced number of viral copies per cell made it more likely that infected cells would survive the infection long enough to produce virus particles themselves.

Viruses that can kill cells, such as HIV, must balance the need to make more of themselves against the speed that they kill their host cell to maximize the number of infected cells. If transmission between cells is too effective and too many virus particles are delivered to the new cell, the virus may not manage to infect new hosts before killing the old ones. These findings highlight this delicate balance. They also indicate a potential issue in using drugs to treat partially resistant virus strains. Without care, these treatments could increase the number of infected cells in the body, potentially worsening the effects of living with HIV.

DOI: https://doi.org/10.7554/eLife.30134.002

cells by initiating a cellular defence program involving the activation of caspase 1 (*Doitsh et al., 2010*; *Doitsh et al., 2014*; *Galloway et al., 2015*). HIV proteins Tat and Env have also been shown to lead to cell death of infected cells through CD95-mediated apoptosis following T-cell activation (*Banda et al., 1992*; *Westendorp et al., 1995a*; *Westendorp et al., 1995b1995*). Using SIV infection, it has been shown that damage to lymph nodes due to chronic immune activation leads to an environment less conducive to T-cell survival (*Zeng et al., 2012*). Finally, double strand breaks in the host DNA caused by integration of the reverse transcribed virus results in cell death by the DNA-PK-mediated activation of the p53 response (*Cooper et al., 2013*).

The lymph node environment is conducive to HIV infection due to: (1) presence of infectable cells (*Deleage et al., 2016*; *Embretson et al., 1993*; *Tenner-Racz et al., 1998*); (2) proximity of cells to each other and lack of flow which should enable cell-to-cell HIV spread (*Baxter et al., 2014*; *Dale et al., 2011*; *Groot et al., 2008*; *Groppelli et al., 2015*; *Gummuluru et al., 2002*; *Hübner et al., 2009*; *Jolly et al., 2004*; *Jolly et al., 2011*; *Münch et al., 2007*; *Sherer et al., 2007*; *Sourisseau et al., 2007*; *Sowinski et al., 2008*); (3) decreased penetration of antiretroviral therapy (ART) (*Fletcher et al., 2014a*). Multiple infections per cell have been reported in cell-to-cell spread of HIV (*Baxter et al., 2014*; *Boullé et al., 2016*; *Dang et al., 2004*; *Del Portillo et al., 2011*; *Dixit and Perelson, 2004*; *Duncan et al., 2013*; *Law et al., 2016*; *Reh et al., 2015*; *Russell et al., 2013*; *Sigal et al., 2011*; *Zhong et al., 2013*). In this mode of HIV transmission, an interaction between the infected donor cell and the uninfected target results in directed transmission of large numbers of virions (*Baxter et al., 2014*; *Groppelli et al., 2015*; *Hübner et al., 2009*; *Sowinski et al., 2008*). This is in contrast to cell-free infection, where free-floating virus finds target

cells through diffusion. Both modes occur simultaneously when infected donor cells are cocultured with targets. However, the cell-to-cell route is thought to be the main cause of multiple infections per cell (*Hübner et al., 2009*). In the lymph nodes, several studies showed multiple infections (*Gratton et al., 2000*; *Jung et al., 2002*; *Law et al., 2016*) while another study did not (*Josefsson et al., 2013*). One explanation for the divergent results is that different cell subsets are infected to different degrees. For example, T cells were shown not to be multiply infected in the peripheral blood compartment (*Josefsson et al., 2011*). However, more recent work investigating markers associated with HIV latency in the face of ART found that the average number of HIV DNA copies per cell is greater than one in 3 out of 12 individuals. This occurred in the face of ART in the CD3-positive, CD32a high CD4 T-cell subset (*Descours et al., 2017*). In the absence of suppressive ART, it would be expected that the number of HIV DNA copies per cell would be higher.

Multiple viral integration attempts per cell may increase the probability of death. One consequence of HIV-mediated death may be that attenuation of infection may increase viral replication by increasing the number of live targets. Indeed, it has been suggested that more attenuated HIV strains result in more successful infections in terms of the ability of the virus to replicate in the infected individual (*Ariën et al., 2005*; *Nowak and May, 2000*; *Payne et al., 2014*; *Quiñones-Mateu and Arts, 2006*; *Wodarz and Levy, 2007*).

Here, we experimentally examined the effect of attenuating cell-to-cell spread by using HIV inhibitors. We observed that partially inhibiting infection with drug or antibody resulted in an increase in the number of live infected cells in both a cell line and in lymph node cells. This is, to our knowledge, the first experimental demonstration at the cellular level that attenuation of HIV infection can result in an increase in live infected cells under specific infection conditions.

## Results

We introduce a model of infection where each donor to target transmission leads to an infection probability $r$ and death probability $q$ per infection attempt. In our experimental system, one infection attempt is measured as one HIV DNA copy, whether integrated or unintegrated. The probability of successful infection of a target cell given $n$ infection attempts is $1-(1-r)^n$ (*Sigal et al., 2011*). We define $L_n$ as the probability of a cell to survive infection in the face of $n$ infection attempts. Assuming infection attempts act independently, $L_n=(1-q)^n$. The probability of a cell to be infected and not die after it has been exposed to $n$ infection attempts is therefore:

$$P_n = (1-(1-r)^n)(1-q)^n \tag{1}$$

This model makes several simplifying assumptions: (1) all infection attempts have equal probabilities to infect targets. (2) The probability for a cell to die from each transmission is equal between transmissions. (3) Infection attempts act independently, and productive infection and death are independent events. In this model, $r$ and $q$ capture the probabilities for a cell to be infected or die post-reverse transcription. For example, mutations which reduce viral fitness by decreasing the probability of HIV to integrate would reduce $r$, while mutations which reduce the probability of successful reverse transcription would reduce $n$.

If the number of infection attempts $n$ is Poisson distributed with mean $\lambda$, the probability for a cell to be infected is $1-e^{-r\lambda}$ and the probability of a cell to live is $L_n = e^{-q\lambda}$ (see **Supplementary file 1** for parameters and definitions). As derived in Appendix 1, the probability that a cell is productively infected will be:

$$P_\lambda = e^{-\lambda q}\left(1 - e^{-\lambda r(1-q)}\right) \tag{2}$$

Since antiretroviral drugs lead to a reduction in the number of infection attempts by, for example, decreasing the probability of reverse transcription in the case of reverse transcriptase inhibitors, we introduced a drug strength value $d$, where $d = 1$ in the absence of drug and $d > 1$ in the presence of drug. In the presence of drug, $\lambda$ is decreased to $\lambda/d$. The drug therefore tunes $\lambda$, and if the antiretroviral regimen is fully suppressive, $\lambda/d$ is expected to be below what is required for ongoing replication. The probability of a cell to be infected and live given drug strength $d$ is therefore:

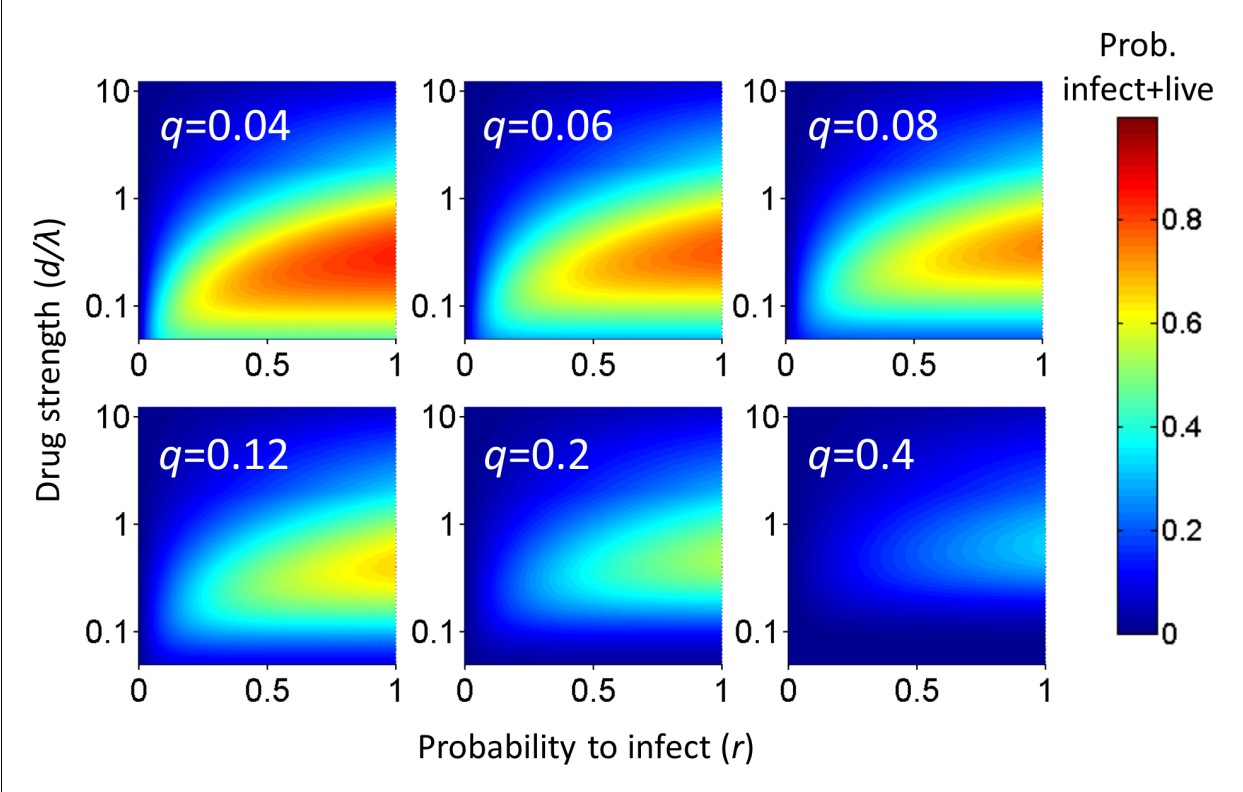

**Figure 1.** Probability for a cell to be infected and live as a function of inhibitor. Probability for a cell to be infected and live was calculated for 20 infection attempts ($\lambda$) and represented as a heat map. Drug strength ($d/\lambda$) is on the y-axis, and the probability per infection attempt to infect ($r$) is on the x-axis. Each plot is the calculation for one value of the probability per infection attempt to die ($q$) denoted in white in the top left corner.
DOI: https://doi.org/10.7554/eLife.30134.003

$$P_{\lambda/d} = e^{-\lambda q/d}\left(1 - e^{-\lambda r(1-q)/d}\right) \qquad (3)$$

Analysis of the probability of a cell to survive and be infected as a function of $r$ and $q$ shows that at each drug strength $d/\lambda$, $P_\lambda$ increases as the probability of infection $r$ increases (*Figure 1*). Hence, the value of $r$ strongly influences the amplitude of $P_\lambda$. How $P_\lambda$ behaves when drug strength $d/\lambda$ increases depends on the parameter values of $r$ and $q$. A subset of parameter values results in a peak in the number of infected cells at intermediate $d/\lambda$, decreasing as drug strength increases further (*Figure 1*). We refer to such a peak in infected numbers as an infection optimum. As $q$ increases, the cost of multiple infections per cell increases, and the infection optimum shifts to higher $d/\lambda$ values. A fall from the infection optimum at decreasing $d/\lambda$ is driven by increasing cell death as a result of increasing infection attempts per cell. This slope is therefore shallower, and peaks broader, at low $q$ values (*Figure 1*).

Our model assumes that cellular infection and death due to an HIV infection attempt are independent processes. This is based on observations that support a role for cell death as a cellular defence mechanism which may occur before productive infection, such as programmed cell death triggered by HIV integration induced DNA damage (*Cooper et al., 2013*). An alternative model is that HIV-mediated cell death depends on productive infection. This would be consistent with cell death due to, for example, expression of viral proteins (*Westendorp et al., 1995b1995*). Since the concentration of viral proteins may also scale with the number of infections per cell, we derived the mathematical model for such a process in the supplementary mathematical analysis. The models are equivalent, showing that independence of cell death and infection is not a necessary condition for an infection optimum to occur in the presence of inhibitor.

Given that an infection optimum is dependent on parameter values, we next examined whether these parameter values occur experimentally in HIV infection. We therefore first tested for an infection optimum in the RevCEM cell line engineered to express GFP upon HIV Rev protein expression (*Wu et al., 2007*). We subcloned the cell line to maximize the frequency of GFP-positive cells upon infection (*Boullé et al., 2016*). We needed to detect the number of infection attempts per cell $\lambda$. To estimate this, we used PCR to detect the number of reverse transcribed copies of viral DNA in the cell by splitting each individual infected cell over multiple wells. We then detected the number of wells with HIV DNA by PCR amplification of the reverse transcriptase gene. Hence, the number of positive wells indicated the minimum number of viral DNA copies per cell, since more than one copy can be contained within the same well (*Josefsson et al., 2011*; *Josefsson et al., 2013*). We first measured the number of viral DNA copies in ACH-2 cells, reported to contain a single inactive HIV integration per genome (*Chun et al., 1997*; *O'Doherty et al., 2002*). We sorted a total of 166 ACH-2 cells at one cell per well into lysis buffer and subdivided single-cell lysates into four wells (*Figure 2—figure supplement 1A*). About one quarter of cells showed a PCR product of the expected size. Cells with more than one HIV copy per cell were very rare and may reflect either errors in cell sorting or dividing cells (*Figure 2—figure supplement 1B*). Similar frequencies were obtained when the ACH-2 cell line was subcloned or split over 10 wells (*Figure 2—figure supplement 1C*). Given that each ACH-2 cell contains one HIV DNA copy, the frequency of detection indicated our detection efficiency per HIV DNA copy.

To investigate the effect of multiple infection attempts per cell, we used coculture infection, where infected (donor) cells are co-incubated with uninfected (target) cells and lead to cell-to-cell spread. We used approximately 2% infected donor cells as our input, and detected the number of HIV DNA copies per cell by flow cytometric sorting of individual GFP-positive cells followed by splitting each cell lysate over 10 wells. Wells were then amplified by PCR and visualized on an agarose gel (*Figure 2A*). We assayed 60 cells and obtained a wide distribution of viral DNA copies per cell, which ranged from 0 to 9 copies (*Figure 2B*). The range of HIV DNA copies per cell fit a Poisson distribution with two means better than either a single mean Gaussian or Poisson distribution. However, the fit of the two mean Poisson distribution did not show two obvious peaks, and instead seemed to fit the data better due to the addition of fit parameters (*Figure 2—figure supplement 2*). Hence we cannot conclude that the distribution is bimodal. We also detected the HIV copy number in 30 GFP-positive cells infected by cell-free HIV. HIV in cell-free form was obtained by filtering supernatant from HIV producing cells to exclude cells or cell fragments, then infecting target cells with the filtered virus. Infection with this virus is defined here as cell-free infection. In this case, we detected either zero or one HIV copy per cell (*Figure 2B* inset). The frequency of single HIV DNA copies was 0.23, identical to the measured result in the ACH-2 cell line. We computationally corrected the detected number of DNA copies in coculture infection for the sensitivity of our PCR reaction as determined by the ACH-2 results (Materials and methods). On average we obtained $15 \pm 3$ copies per cell after correction.

To tune $\lambda$, we added the HIV reverse transcriptase inhibitor efavirenz (EFV) to infections. To calculate $d$, we used cell-free infection (*Figure 2C*, see *Figure 2—figure supplement 3* for logarithmic y-axis plot), which as verified above, results in single HIV copies per cell. For cell-free infection, we approximate $d = 1/Tx$, where $Tx$ is defined as the number of infected cells with drug divided by the number of infected cells without drug with single infection attempts (see Materials and methods and [(*Sigal et al., 2011*)]). This is equivalent to $1-\varepsilon$ in a commonly used model describing the effect of inhibitors on infection. In this model, $\varepsilon$ is drug effectiveness, with the 50% inhibitory drug concentration ($IC_{50}$) and the Hill coefficient for drug action as parameter values (*Canini and Perelson, 2014*; *Shen et al., 2008*). We fit the observed response of infection to EFV using this approach to estimate $d$ across a range of EFV concentrations. Fit of the model to the cell-free data using wild type, EFV-sensitive HIV showed a monotonic decrease with $IC_{50}$ = 2.9 nM and Hill coefficient of 2.1 (*Figure 2C*, black line).

We next dialed in EFV to tune $\lambda/d$ in coculture infection. To obtain the number of infected target cells, and specifically exclude donor cells or donor-target cell fusions, target cells were marked by the expression of mCherry. Donor cells were stained with the vital stain Cell Trace Far Red (CTFR). The concentration of live infected cells was determined after 2 days in coculture with infected donors. Live infected cells were identified based on the absence of cell death indicator dye DAPI fluorescence, and presence of GFP. The input of infected donor cells was excluded from the count

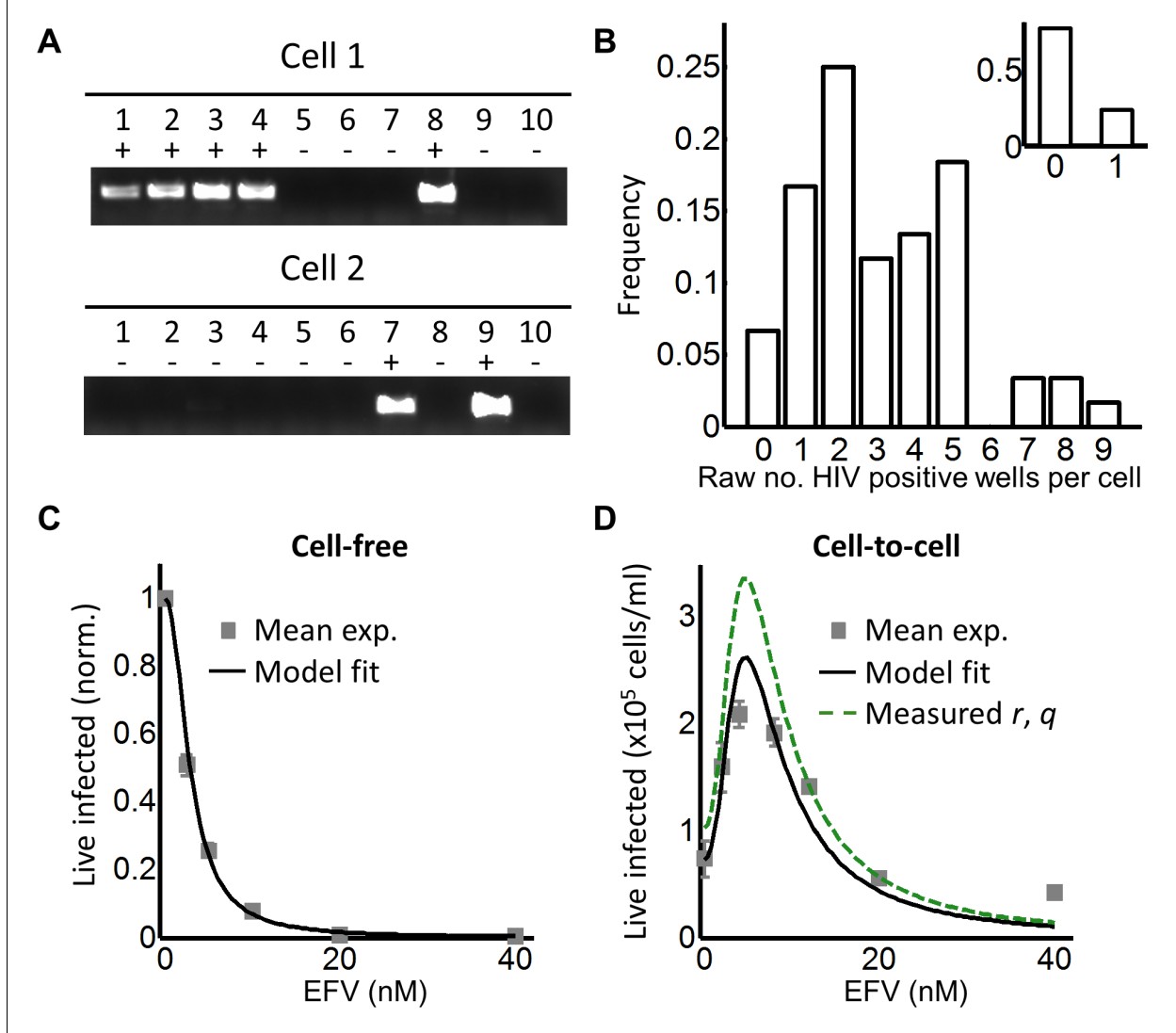

**Figure 2.** Partial inhibition increases the number of live infected cells. (**A**) To quantify HIV DNA copy number per cell, GFP-positive cells were sorted into individual wells and lysed. Each lysate was subdivided into 10 wells and PCR performed to detect HIV DNA, with the sum of positive wells being the raw HIV copy number for that cell. (**B**) Histogram of raw HIV DNA copies per cell in coculture infection (n = 60 cells, four independent experiments). Inset shows raw HIV DNA copies per cell in cell-free infection (n = 30, two independent experiments) (**C**) Number of live infected cells normalized by maximum number of live infected cells in cell-free infection with EFV. Black line is best-fit for EFV suppression of cell-free infection ($IC_{50}$ = 2.9 nM, h = 2.1). Means and standard errors for three independent experiments. (**D**) Number of live infected cells/ml 2 days post-infection resulting from coculture infection of $10^6$ cells/ml in the presence of EFV. Means and standard errors for three independent experiments. Black line is best-fit of *Equation (3)* with r = 0.22 and q = 0.17. Dashed green line is the result of *Equation (3)* with experimentally measured r = 0.28, q = 0.15.

DOI: https://doi.org/10.7554/eLife.30134.004

The following figure supplements are available for figure 2:

**Figure supplement 1.** Detected integrations in ACH-2 cells.

DOI: https://doi.org/10.7554/eLife.30134.005

**Figure supplement 2.** Fitting of different distributions to the frequency of HIV DNA copies per cell.

DOI: https://doi.org/10.7554/eLife.30134.006

**Figure supplement 3.** The number of live infected cells in cell-free infection with wild-type HIV.

DOI: https://doi.org/10.7554/eLife.30134.007

**Figure supplement 4.** Gating strategy for coculture infection with wild type HIV.

DOI: https://doi.org/10.7554/eLife.30134.008

**Figure supplement 5.** Experimental measurement of r and q.

DOI: https://doi.org/10.7554/eLife.30134.009

*Figure 2 continued on next page*

*Figure 2 continued*

**Figure supplement 6.** Time-lapse microscopy of HIV infection in the absence and presence of EFV.

DOI: https://doi.org/10.7554/eLife.30134.010

of infected cells based on the absence of mCherry fluorescence. Donor-target cell fusions were excluded by excluding CTFR-positive cells (see *Figure 2—figure supplement 4* for gating strategy).

While the percent of infected cells was reduced with drug, the concentration of live infected cells increased (*Figure 2—figure supplement 4*). We observed a peak in the number of live infected target cells at 4 nM EFV (*Figure 2D*). We then fit the number of live infected cells using *Equation (3)*, where $P_\lambda$ was multiplied by the input number of target cells per ml ($10^6$ cells/ml) to obtain the predicted number of live infected cells per ml of culture. This was done to constrain $r$ in the model, which strongly determines the amplitude of $P_{\lambda/d}$ as described above. *Equation (3)* best fit the behaviour of infection when $r = 0.22$ *and* $q = 0.17$, resulting in a peak at 4.8 nM EFV (*Figure 2D*, black line). Hence an infection optimum is present in the cell line infection system.

In order to determine whether the fitted $r$ and $q$ values were within a reasonable range, we measured these values experimentally. To measure $r$, we infected with cell-free HIV to avoid the broad distribution of HIV copy numbers observed in cell-to-cell spread, and determined the fraction of live infected cells $P_\lambda$ (*Figure 2—figure supplement 5A*). We then determined the mean number of HIV copies per cell $\lambda$ for the same set of experiments corrected by the efficiency of detection (*Figure 2—figure supplement 5B*). The parameter $r$ was calculated as $-ln(1-P_\lambda)/\lambda$ (*Supplementary file 2*). To measure $q$, we blocked cell division using serum starvation to measure differences in cell concentration due to cell death only, and not due to proliferation of uninfected cells (*Figure 2—figure supplement 5C*). We then infected with cell-free HIV and measured $L_\lambda$, defined as the fraction of live cells remaining upon infection with $\lambda$ HIV DNA copies relative to infection blocked with EFV (see below). To specifically detect the decrease in live cells as a result of events downstream of reverse transcription, we compared infected cells to cells exposed to the same virus concentration but treated with 40 nM EFV, a drug concentration where infection by cell-free virus is negligible (*Figure 2—figure supplement 3*). $q$ was then calculated as $-ln(L_\lambda)/\lambda$, where $L_\lambda$ was the probability of a cell to live given transmission with $\lambda$ copies (*Supplementary file 2*). Measured $r$ and $q$ values were $0.28 \pm 0.08$ and $0.15 \pm 0.07$ (mean $\pm$standard deviation), respectively. The solution to *Equation (3)* using these values showed similar behavior to the solution with the fitted values for wild-type HIV infection, indicating that the fitted values gave a reasonable approximation of the behavior of the system (*Figure 2D*, dashed green line).

In order to investigate the dynamics of cell depletion due to cell-to-cell HIV spread and its modulation by the addition of an inhibitor, we performed time-lapse microscopy over a two day infection window. While infection parameters were different due to the constraints of visualizing cells (Materials and methods), the general trend from the data was deterioration in the number of live cells in the time-lapse culture starting at 1 day post-infection when no drug was added. The deterioration in live cell numbers was averted by the addition of EFV (*Figure 2—figure supplement 6*).

We next investigated whether an infection optimum occurs with EFV-resistant HIV. To derive the resistant mutant, we cultured wild-type HIV in our reporter cell line in the presence of EFV. We obtained the L100I partially resistant mutant. We then replaced the reverse transcriptase of the wild-type molecular clone with the mutant reverse transcriptase gene (Materials and methods). We derived $d_{mut}$ for the L100I mutant using cell-free mutant infection (*Figure 3A*, see *Figure 3—figure supplement 1* for logarithmic y-axis plot). The L100I mutant was found to have an $IC_{50} = 29$ nM EFV and a Hill coefficient of 2.0 (*Figure 3A*, black line).

We next performed coculture infection (see *Figure 3—figure supplement 2* for gating strategy). Similarly to wild-type HIV coculture infection, there was a peak in the number of live infected target cells for the L100I mutant infection. However, the peak in live infected cells was shifted to 40 nM EFV (*Figure 3B*). Fits were obtained to *Equation (3)* using $d_{mut}$ values and $\lambda$ measured for wild-type infection. The fits recapitulated the experimental results when $r = 0.29$ and $q = 0.13$, with a fitted peak at 45 nM EFV (*Figure 3B*, black line). The solution to *Equation (3)* using the measured values for $r$ and $q$ showed a similar pattern to that obtained with the fitted values (*Figure 3B*, dashed green line). We note that both wild type and mutant coculture infection has data points above the fit line at the highest drug concentrations. This may be a limitation of our model at drug values much higher

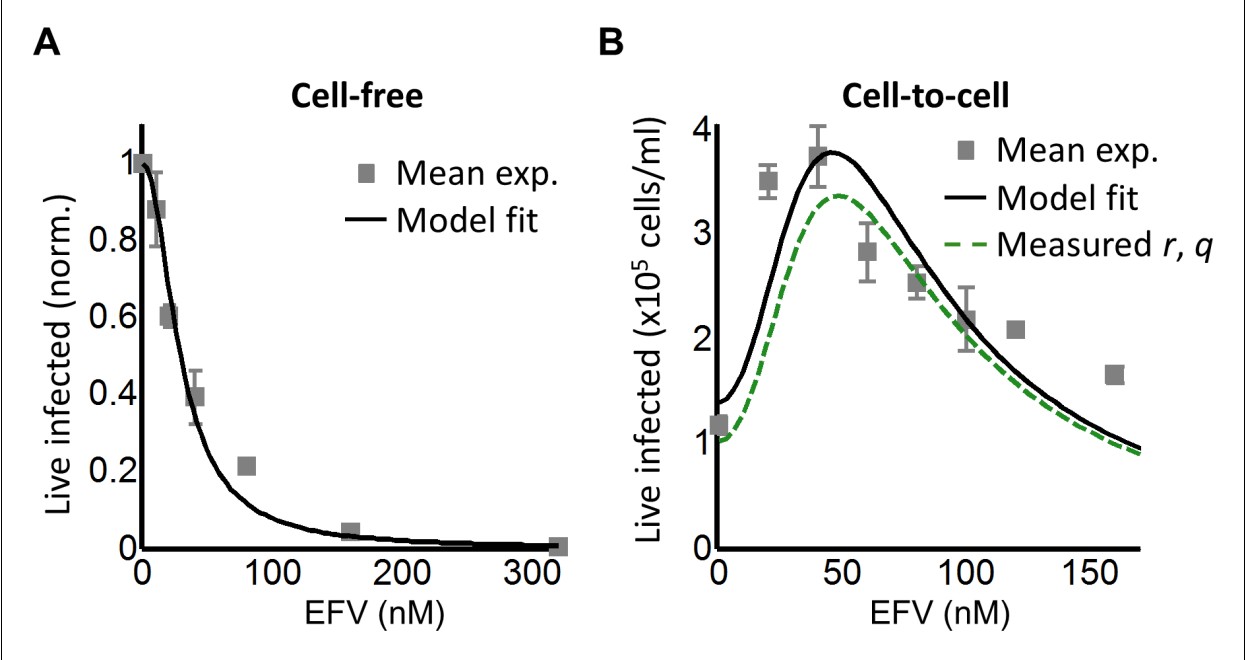

**Figure 3.** Partial inhibition of the EFV-resistant L100I mutant shifts the peak of live infected cells to higher EFV concentrations. (A) The number of live infected cells normalized by the maximum number of live infected cells in cell-free infection as a function of EFV for the L100I mutant. Black line is best-fit for EFV suppression of cell-free infection (IC$_{50}$ = 29 nM, h = 2.0). Shown are means and standard errors for three independent experiments. (B) Number of live infected cells/ml 2 days post-infection resulting from coculture infection of $10^6$ cells/ml in the presence of EFV. Means and standard errors for three independent experiments. Black line is best-fit of *Equation (3)* with r = 0.29 and q = 0.13. Dashed green line is the result of *Equation (3)* with the experimentally measured r = 0.28 and q = 0.15 for wild-type HIV infection.

DOI: https://doi.org/10.7554/eLife.30134.011

The following figure supplements are available for figure 3:

**Figure supplement 1.** The number of live infected cells in cell-free infection with the L100I mutant.

DOI: https://doi.org/10.7554/eLife.30134.012

**Figure supplement 2.** Gating strategy for coculture infection with mutant HIV.

DOI: https://doi.org/10.7554/eLife.30134.013

than observed at the infection optimum. In this range of drug values, our model predicts a more pronounced decline in the number of infected cells than is observed experimentally.

In order to examine whether a peak in live infected targets can be obtained with an unrelated inhibitor, we used the HIV neutralizing antibody b12. This antibody is effective against cell-to-cell spread of HIV (*Baxter et al., 2014*; *Reh et al., 2015*). We obtained a peak in live infected cells at 5 ug/ml b12 (*Figure 4*). The b12 concentration that resulted in a peak number of live infected cells was the same for wild-type virus and the L100I mutant, showing that L100I mutant fitness gain was EFV specific. In contrast, cell-free infection in the face of b12 showed a sharp and monotonic drop in live infected cells for both wild type and mutant virus (*Figure 4—figure supplement 1*).

While the RevCEM cell line is a useful tool to illustrate the principles governing the formation of an infection optimum, the sensitivity of such an optimum to parameter values would make its presence in primary HIV target cells speculative. We therefore investigated whether a fitness optimum occurs in primary human lymph node cells, the anatomical site which would be most likely to have a high force of infection. We derived human lymph nodes from HIV-negative individuals from indicated lung resections (*Supplementary file 3*), cellularized the lymph node tissue using mechanical separation, and infected the resulting lymph node cells with HIV. A fraction of the cells was infected by cell-free virus and used as infected donor cells. We added these to uninfected target cells from the same lymph node to test coculture infection, and detected the number of live infected cells 4 days post-infection with the L100I EFV-resistant mutant in the face of EFV. We detected the number of

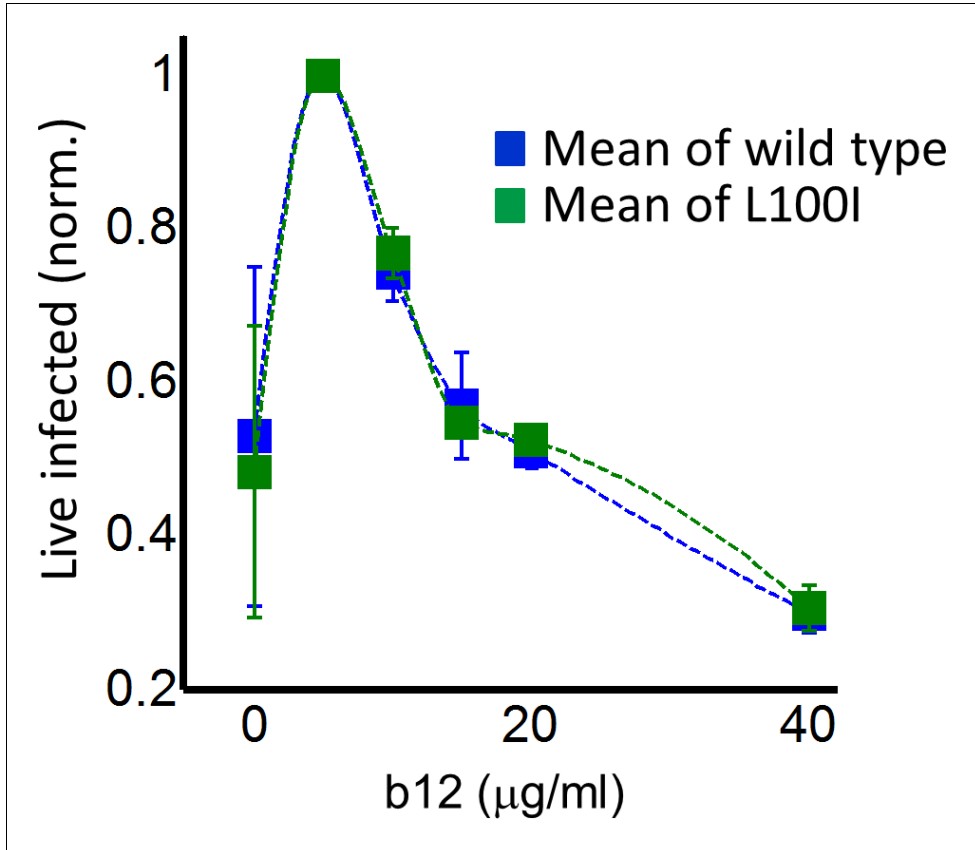

**Figure 4.** Partial inhibition of coculture infection with neutralizing antibody results in higher numbers of live infected cells. Shown are the numbers of live infected cells normalized by the maximum number of live infected cells in coculture infection as a function of b12 antibody concentration. Infection was by either EFV-sensitive HIV (blue) or the L100I EFV-resistant mutant (green). Dashed lines are a guide to the eye. Shown are means and standard errors for three independent experiments.

DOI: https://doi.org/10.7554/eLife.30134.014

The following figure supplement is available for figure 4:

**Figure supplement 1.** The number of live infected cells with cell-free infection in the face of neutralizing antibody b12.

DOI: https://doi.org/10.7554/eLife.30134.015

live infected cells by the exclusion of dead cells with the fixable death detection dye eFluor660 followed by single cell staining for HIV Gag using anti-p24 antibody (*Figure 5A*).

In each of the lymph nodes tested, we observed a peak in live infected cells at intermediate EFV concentrations. Lymph node cells from participant 205 showed a peak of live infected cells at 100 nM EFV (*Figure 5A*, first row). The infection optimum in the lymph node cells of study participant 257 was visible as a plateau between 50 and 200 nM EFV. In the presence of EFV, there was a decrease in the fraction of dead cells that was offset by a similar increase in the fraction of live infected cells for lymph nodes from all participants. There were more overall detectable cells with EFV, resulting in differences in the absolute concentrations of live infected cells being larger than the differences in the fractions of live infected cells between EFV and non-drug-treated cells (right two columns in *Figure 5A*, with absolute number of live infected cells shown in parentheses in the flow cytometry plots). This is most likely due to cells which died early becoming fragments and so being excluded from the total population in the absence of EFV. Peaks in the number of live infected cells in the face of drug may be specific to lymph node derived cells. Cell-to-cell infection of peripheral blood mononuclear cells (PBMC) with wild-type HIV showed a slight peak at a very low EFV concentration in cells from one blood donor, which was not repeated in cells from two other donors (*Figure 5—figure supplement 1*).

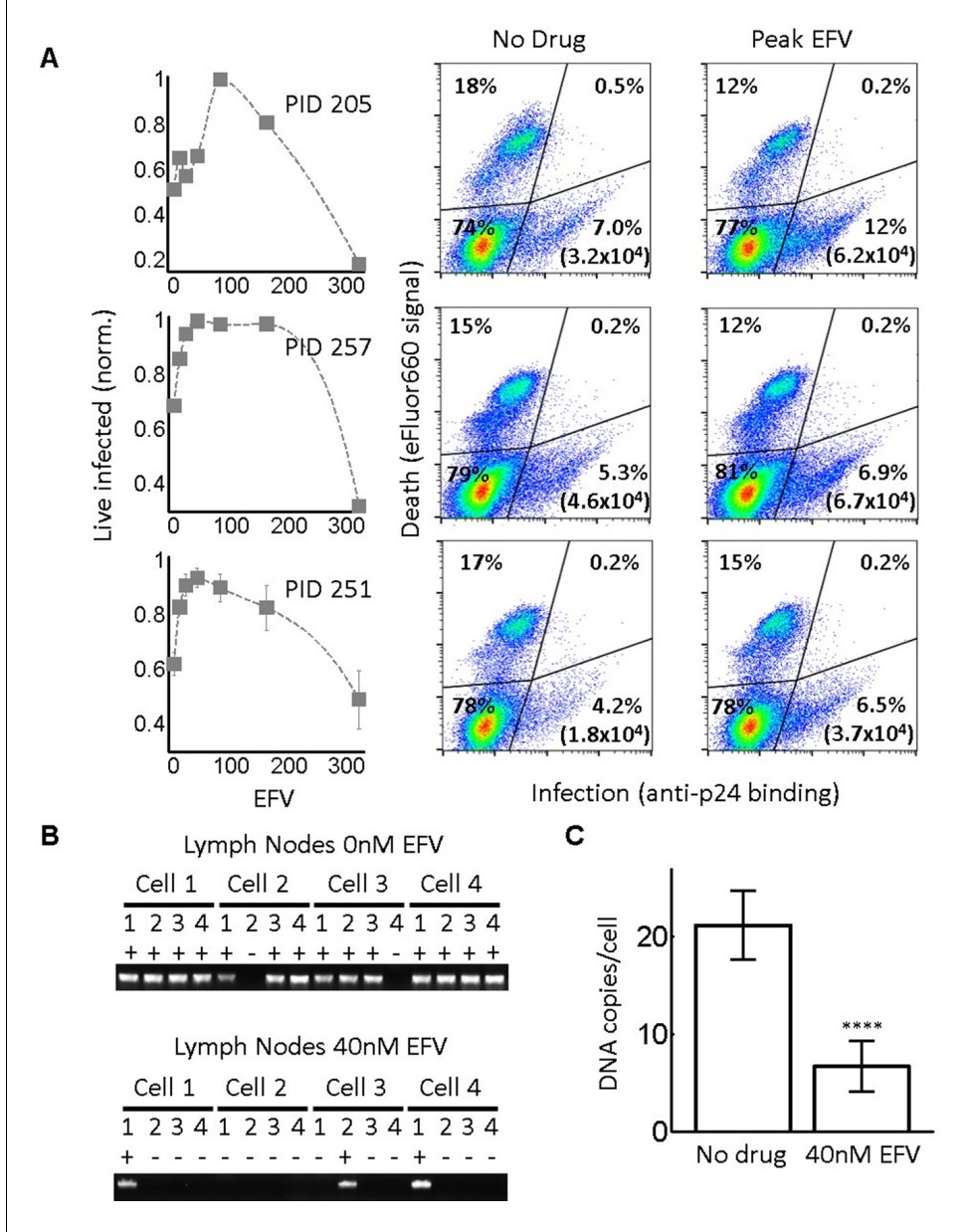

**Figure 5.** Infection optimum with EFV in lymph node cells. (**A**) Number of live infected cells as a function of EFV. Each row shows in vitro infected lymph node cells from one participant. Left column is the number of live infected cells normalized by maximum number of live infected cells in coculture infection. Middle and right columns are flow cytometry dot plots of infection without drug and at the infection optimum, with HIV p24 on x-axis and death detection by eFluor660 on y-axis. Infected live cells are bottom right. Number in brackets represents live infected cell density per ml. PID, participant identification number. For PID205 and 257, cells were sufficient for one experiment. For PID251, means and standard errors for three independent experiments are shown. Dashed lines are a guide to the eye. (**B**) HIV DNA copy number per cell was quantified by sorting fixed p24-positive cells from PID251 into individual wells. Cells were lysed and de-crosslinking performed. Each lysate was divided into four wells and PCR performed to detect HIV DNA. First row is representative of cells with no drug, second row is representative of 40 nM EFV. (**C**) Mean and standard error of the number of HIV DNA copies without drug and with 40 nM EFV after assay sensitivity correction. N = 56 cells from three independent experiments for each condition. ****$p=4\times10^{-9}$, two tailed t-test.

DOI: https://doi.org/10.7554/eLife.30134.016

The following figure supplements are available for figure 5:

*Figure 5 continued on next page*

*Figure 5 continued*

**Figure supplement 1.** Cell-to-cell infection of peripheral blood mononuclear cells in the presence of EFV does not lead to a discernable peak in live infected cells.

DOI: https://doi.org/10.7554/eLife.30134.017

**Figure supplement 2.** Raw HIV DNA copy numbers per lymph node cell.

DOI: https://doi.org/10.7554/eLife.30134.018

We used a lymph node from study participant 251, where we obtained more cells, to examine the number of HIV DNA copies per cell. Cells from this lymph node showed an infection optimum at 50 nM EFV (*Figure 5A*, third row). To detect the effect of EFV on integrations per cell, we sorted single cells based on p24-positive signal, de-crosslinked to remove the fixative (Materials and methods), then divided each cell lysate into four wells. Using fewer wells saved reagents without changing sensitivity, as demonstrated in the ACH-2 cell line (*Figure 2—figure supplement 1C*). We detected HIV DNA copies by PCR 2 days post-infection. We observed multiple DNA copies in EFV-untreated lymph node cells. The number of copies decreased with EFV (*Figure 5B*). We corrected for sensitivity of detection as quantified in ACH-2 cells (Materials and methods). The corrected numbers were 21 HIV DNA copies with no drug, and five copies in the presence of EFV at the infection optimum (*Figure 5C*, see *Figure 5—figure supplement 2* for histograms of raw HIV DNA copy numbers per cell). Hence, the decrease in the number of copies still results in sufficient copies to infect the cell.

Since L100I does not often occur in the absence of other drug resistance mutations according to the Stanford HIV Drug Resistance Database (*Rhee et al., 2003*), we repeated the experiment with the K103N mutant, a frequently observed mutation in virologic failure with a higher level of resistance to EFV relative to the L100I mutant. We used cell-free infection to obtain drug inhibition per virion at each level of EFV, which we denote $d_{103}$ (*Figure 6A*, see *Figure 6—figure supplement 1* for logarithmic y-axis plot). The fits showed a monotonic decrease with $IC_{50}$ = 26.0 nM and Hill coefficient of 1.5 (*Figure 6A*, black line). We then proceeded to use the K103N mutant in coculture infection, using cells from two different lymph nodes in different experiments (see *Figure 6—figure supplement 2* for results of individual experiments). We observed an infection optimum with EFV in lymph node cells. The peak in the number of live infected cells in the presence of drug was between 80 and 160 nM EFV (*Figure 6B*). We fit the experimental data with *Equation (3)* using $d_{103}$ values and the number of DNA copies in the absence of drug measured for L100I infection. We did not calculate the predicted number of infected cells for $P_{\lambda/d}$ values since the lymph node is a complex environment containing different cell subsets (*Sallusto et al., 1999*) and the number of infectable target cells at the start of infection is difficult to determine. Hence, we normalized both the experimental number of live infected cells and the $P_{\lambda/d}$ values from *Equation (3)* to the maximum value in each case. The fits recapitulated the experimental results when $r = 0.91$ and $q = 0.15$, with a fitted peak at 90 nM EFV (*Figure 6B*, black line). The $q$ value matched the measured result in the cell line, while the $r$ value was much higher. However, the fitted $r$ value in this case is not expected to be accurate since we were unable to constrain it with the number of infected cells relative to the starting number of target cells.

To examine if the observed peak in live cells may be due to EFV alone, we measured cell viability in lymph node cells from one of the study participants used in the above experiment as a function of EFV without infection. No clear dependence on EFV in the absence of infection was detected (*Figure 6—figure supplement 3*).

## Discussion

The optimal virulence concept in ecology proposes that virulence needs to be balanced against host survival for optimal pathogen spread (*Bonhoeffer et al., 1996*; *Bonhoeffer and Nowak, 1994*; *Gandon et al., 2001*; *Jensen et al., 2006*). At the cellular level, this implies that the number of successfully infected cells may increase when infection virulence is reduced. The current study is, to our knowledge, the first to address this question experimentally at the level of individual cells infected with HIV.

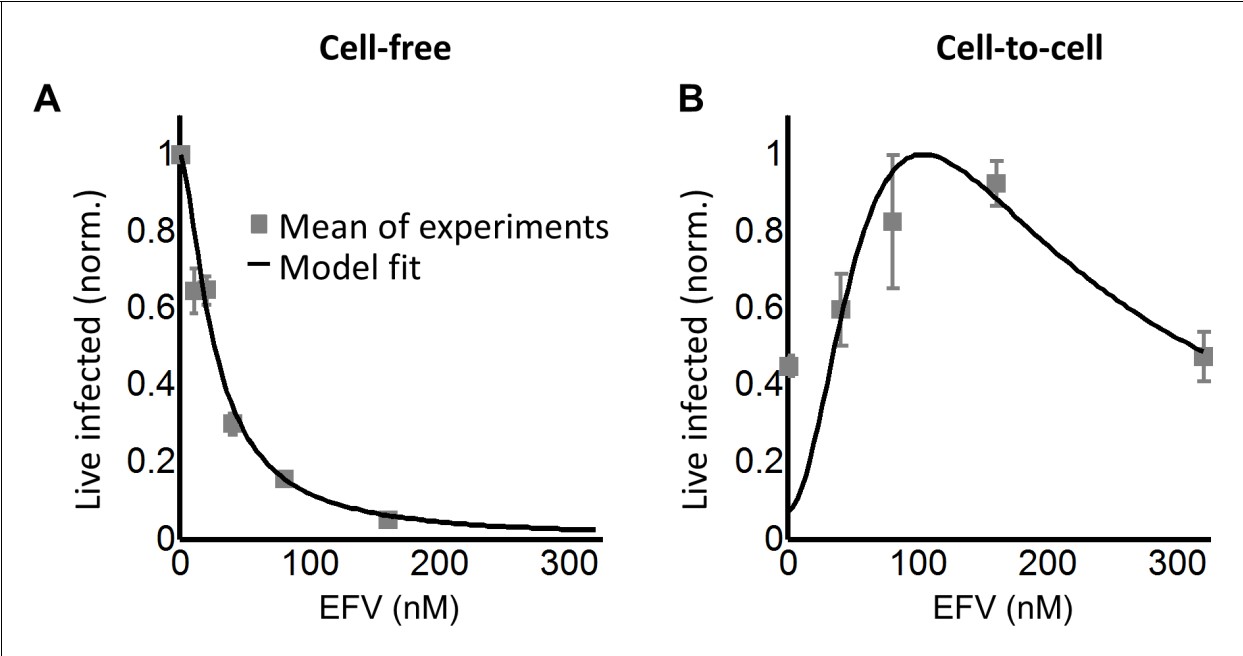

**Figure 6.** Infection with the K103N mutant shows an infection optimum at clinically observed lymph node EFV concentrations. (**A**) The number of live infected cells normalized by the maximum number of live infected cells in cell-free infection as a function of EFV for the K103N mutant. Black line is best-fit for EFV suppression of cell-free infection ($IC_{50}$ = 26 nM, h = 1.5). Shown are means and standard errors for two independent experiments using cells from PID251. (**B**) The number of live infected cells normalized by the maximum number of live infected cells in coculture infection as a function of EFV for the K103N mutant. Black line represents best-fit model for the effect of EFV on coculture infection according to *Equation (3)*, with *d* values calculated based on the cell-free infection data for the K103N mutant, and the mean number of HIV DNA copies in the absence of drug determined for L100I. The fits recapitulated the experimental results when *r* = 0.91 and *q* = 0.15. Shown are means and standard errors for three independent experiments. There were sufficient lymph node cells from PID251 for two of the three experiments, and the third experiment was performed with lymph node cells from PID274.

DOI: https://doi.org/10.7554/eLife.30134.019

The following figure supplements are available for figure 6:

**Figure supplement 1.** The number of live infected cells in cell-free infection with the K103N mutant.
DOI: https://doi.org/10.7554/eLife.30134.020
**Figure supplement 2.** Response of K103N coculture infection to EFV in individual experiments.
DOI: https://doi.org/10.7554/eLife.30134.021
**Figure supplement 3.** Effect of EFV on viability of uninfected lymph node cells from PID274.
DOI: https://doi.org/10.7554/eLife.30134.022

Using a model where cells are infected and die in a probabilistic way, we found that there were two possible outcomes of partially inhibiting infection. In the case where cells were infected by single infection attempts, inhibition always led to a decline in the number of live infected cells, since inhibition reduced the number of infections per cell from one to zero. In contrast, in the case of multiple infection attempts per cell, the possibility existed that inhibition reduced the number of integrating HIV DNA copies, without extinguishing infection of the cell completely. If each HIV DNA copy increases the probability of cell death, reducing the number of HIV DNA copies without eliminating infection should lead to an increased probability of infected cell survival. This would consequently lead to an increase in the number of live infected cells.

We investigated the outcome of partial inhibition of infection in both a cell line and primary lymph node cells. In both systems, we observed that there was a peak in live infected cell number at intermediate inhibitor concentrations. This correlated to a decreased number of viral DNA copies per cell. Further increasing inhibitor concentration led to a decline in live infected cell numbers, and infecting with EFV resistant mutants shifted the peak in live infected number to higher EFV concentrations. Our model as described by *Equation (3)* reproduced the essential behaviour of the experimental results. Construction of the model assumed independence of productive infection and cell

death. However, as shown in the Appendix 1, an equivalent model can be constructed assuming a dependence of cell death on infection. Neither model accurately captures infection dynamics at high-drug concentrations, away from the infection optimum. In this range, where the number of infection attempts per cell is much lower than 1, infection declined more slowly with drug than predicted. The model can be further refined using a distribution for the number of DNA copies per cell. Moreover, the probability of death per HIV DNA copy we denote $q$ may be dependent on how many infection attempts preceded the current infection attempt, and the model can be improved by measuring this dependence.

Physiologically, an infection optimum in the face of an antiretroviral drug may be important in HIV infection of lymph node cells and may be less pronounced in cells from peripheral blood. We used EFV in our study since it is a common component of first line antiretroviral therapy, with frequent drug resistance mutations. However, the infection optimum we describe should occur with other classes of antiretroviral drugs, since all drugs should decrease the multiplicity of infection between cells. In terms of modeling, a future therapy component such as the integrase inhibitor dolutegravir would exert its effect on $r$ and not $\lambda$ in our model. However, the effect is symmetrical since $e^{-(\lambda/d)}{}^r = e^{-\lambda(r/d)}$. The more complex outcome of partial inhibition of infection should also be considered in other infections where multiple pathogens infect one cell and host cell death is a possible outcome (*Mahamed et al., 2017*).

These observations reinforce previous results showing that successful completion of reverse transcription leads to cellular cytotoxicity. In addition to HIV cytotoxicity caused by viral integrations through the mechanism of double strand breaks (*Cooper et al., 2013*), other mechanisms of HIV-induced death are also present, including IFI16-dependent innate immune system sensing of abortive reverse transcripts following non-productive infection of resting T cells (*Doitsh et al., 2014*; *Monroe et al., 2014*). The experiments presented here reflect the effect of partial inhibition on productive infection of HIV target cells, which mostly consist of activated T-cell subsets, not resting T cells. More complex models would be needed to decipher the effect of partial inhibition of HIV infection on resting T-cell numbers and the outcome of this in terms of available T-cell targets in future infection cycles.

This study may have implications for the establishment of viral reservoirs in the context of poorly controlled infections, infections with some degree of drug resistance, or infections where some replication may take place in the face of ART, since infected cell survival is a pre-requisite for long-term persistence. The clinical implications of an infection optimum in the presence of EFV with EFV-sensitive HIV strains are likely to be negligible, since the drug concentrations at which the infection optimum occurs are extremely low. However, for EFV-resistant HIV, the infection optimum shifts to the range of EFV concentrations observed in lymph nodes (~100 nM) (*Fletcher et al., 2014b*), and can be expected to shift to even higher EFV concentrations with more resistant mutants. As EFV has a longer half-life than the other antiretroviral drugs co-formulated with it, it may be the only agent present in partially adherent individuals for substantial periods of time (*Taylor et al., 2007*). Therefore, partial inhibition of HIV infection with EFV may provide a surprising advantage to EFV resistant mutants, and may allow individuals failing therapy to better transmit drug resistant strains.

## Materials and methods

### Ethical statement

Lymph nodes were obtained from the field of surgery of participants undergoing surgery for diagnostic purposes and/or complications of inflammatory lung disease. Informed consent was obtained from each participant, and the study protocol approved by the University of KwaZulu-Natal Institutional Review Board (approval BE024/09). Blood for PBMC was obtained from healthy blood donors under the same study protocol.

### Inhibitors, viruses and cell lines

The following reagents were obtained through the AIDS Research and Reference Reagent Program, National Institute of Allergy and Infectious Diseases, National Institutes of Health: the antiretroviral EFV; RevCEM cells from Y. Wu and J. Marsh; HIV molecular clone pNL4-3 from M. Martin; ACH-2 cells from T. Folks. Cell-free viruses were produced by transfection of HEK293 cells with pNL4-3

using TransIT-LT1 (Mirus, Madison, WI ) or Fugene HD (Roche, Risch-Rotkreuz, Switzerland) transfection reagents. Virus containing supernatant was harvested after 2 days of incubation and filtered through a 0.45 µm filter (Corning, New York, NY). b12 antibody was produced from transfecting HEK293 cells with a b12 expression plasmid (expressed under a CMV promoter on a pHAGE6 lentiviral plasmid backbone, gift from A. Balazs), followed by harvesting of cell supernatant and purification at the California Institute of Technology protein expression core. The number of virus genomes in viral stocks was determined using the RealTime HIV-1 viral load test (Abbott Diagnostics, Santa Clara, CA). For $r$ and $q$ measurement, 0.45 µm filtered cell-free supernatants from infected RevCEM cells were used, to include any secreted factors which may modulate cell-death. The L100I and K103N mutants were evolved by serial passages of wild-type NL4-3 in RevCEM cells in the presence of 20 nM EFV. After 16 days of selection, the reverse transcriptase gene was cloned from the proviral DNA and the mutant reverse transcriptase gene was inserted into the NL4-3 molecular clone. RevCEM clones E7 and G2 used in this study were generated as previously described (Boullé et al., 2016). Briefly, the E7 clone was generated by subcloning RevCEM cells at single-cell density. Surviving clones were subdivided into replicate plates. One of the plates was screened for the fraction of GFP expressing cells upon HIV infection using microscopy, and the clone with the highest fraction of GFP-positive cells was selected. To generate the G2 clone, E7 cells were stably infected with the mCherry gene under the EF-1α promoter on a pHAGE2-based lentiviral vector (gift from A. Balazs), subcloned, and screened for >99% mCherry-positive cells. All cell lines not authenticated, and mycoplasma negative. Cell culture and experiments were performed in complete RPMI 1640 medium supplemented with L-Glutamine, sodium pyruvate, HEPES, non-essential amino acids (Lonza, Basel, Switzerland), and 10% heat-inactivated FBS (GE Healthcare Bio-Sciences, Pittsburgh, PA).

## Primary cells

Lymph node cells were obtained by mechanical separation of lymph nodes and frozen at $5 \times 10^6$ cells/ml in a solution of 90% FBS and 10% DMSO with 2.5 µg/ml Amphotericin B (Lonza). Cells were stored in liquid nitrogen until use, then thawed and resuspended at $10^6$ cells/ml in complete RPMI 1640 medium supplemented with L-Glutamine, sodium pyruvate, HEPES, non-essential amino acids (Lonza), 10% heat-inactivated FBS (Hyclone), and IL-2 at 5 ng/ml (PeproTech). Phytohemagglutinin at 10 µg/ml (Sigma-Aldrich, St Louis, MO) was added to activate cells. PBMCs were isolated by density gradient centrifugation using Histopaque 1077 (Sigma-Aldrich) and cultured at $10^6$ cells/ml in complete RPMI 1640 medium supplemented with L-Glutamine, sodium pyruvate, HEPES, non-essential amino acids (Lonza), 10% heat-inactivated FBS (GE Healthcare Bio-Sciences, Pittsburgh, PA), and IL-2 at 5 ng/ml (PeproTech, Rocky Hill, NJ). Phytohemagglutinin at 10 µg/ml (Sigma-Aldrich) was added to activate cells. For both primary cell types, donor cells for coculture infection were cultured for one day then infected by cell-free virus, while target cells were cultured for three days and infected with either cell-free HIV or infected donor cells.

## Subcloning of ACH-2 cells

Cells from the parental ACH-2 cell line were diluted to 10 cells/ml in conditioned medium, with conditioned medium generated by culturing ACH-2 cells to $10^6$ cells/ml, then filtering through a 0.22 µm filter (Corning). 25 µl of the diluted cell suspension was then distributed to each well of a Greiner µClear 384-well plate (mean of 0.5 cells per well). Clones were cultured for 3 weeks, where each week an additional 25 µl of conditioned medium was added to each well. Clones were detected in 5% of wells and two clones, designated D6 and C3, were randomly chosen and further expanded.

## Infection

For a cell-free infection of RevCEM clones, PBMC and lymph node cells, $10^6$ cells/ml were infected with $2 \times 10^8$ NL4-3 viral copies/ml (~20ng p24 equivalent) for 2 days. For coculture infection, infected cells from the cell-free infection were used as the donors and cocultured with $10^6$ cells/ml target cells. For RevCEM clones, 2% infected donor cells were added to uninfected targets and cocultured for 2 days in tissue culture experiments, and 20% infected donor cells were added to uninfected targets and cocultured for 2 days for time-lapse experiments. For lymph node cells and

PBMCs, a ratio of 1:4 donor to targets cells was used. Infection was over 2 days in PBMC infection and for 4 days for infection of lymph node cells.

## Staining and flow cytometry

To determine the number of live infected cells in reporter cell line experiments, E7 RevCEM reporter cells were infected as above used as donor cells. Prior to co-incubation with target cells, donor cells were stained with CellTrace Far Red (CTFR, Thermo Fisher Scientific, Waltham, MA) at 1 μM and washed according to manufacturer's instructions. The G2 mCherry-positive reporter cells were used as infection targets, and cocultured with 2% infected donor cells for 2 days. The coculture infection was pulsed with 100 ng/ml DAPI (Sigma-Aldrich) immediately before flow cytometry and the number of live infected targets cells was determined by the number of DAPI negative, CTFR negative and mCherry and GFP double positive cells on a FACSAria Fusion machine (BD Biosciences, Sparks, MD) using the 355, 488 and 633 nm laser lines. For cell-free infections where fewer fluorescence channels were used, a pulse of 300 nM of the far-red live cell impermeable dye DRAQ7 (Biolegend, San Diego, CA) immediately before flow cytometry was substituted for DAPI, and live infected cells detected as the number of DRAQ7-negative, GFP-positive cells on a FACSCaliber machine using 488 and 633 nm laser lines. Lymph node cells were resuspended in 1 ml of phosphate buffered saline (PBS) and stained at a 1:1000 dilution of the eFlour660 dye (Thermo Fisher Scientific) according to the manufacturer's instructions. Cells were then fixed and permeabilized using the BD Cytofix/Cyto-perm Fixation/Permeabilization kit (BD Biosciences) according to the manufacturer's instructions. Cells were then stained with anti-p24 FITC conjugated antibody (KC57, Beckman Coulter, Brea, CA). Live infected lymph node cells were detected as the number of eFluor660-negative, p24-positive cells. Cells were acquired with a FACSAriaIII or FACSCaliber machine (BD Biosciences) using 488 and 633 nm laser lines. Results were analysed with FlowJo 10.0.8 software. For single-cell sorting to detect the number of HIV DNA copies per cell, cells were single-cell sorted using 85 micron nozzle in a FACSAriaIII machine. GFP-positive, DRAQ7-negative RevCEM clones were sorted 1 day post-infection into 96 well plates (Biorad, Hercules, CA) containing 30 μl lysis buffer (2.5 μl 0.1M Dithio-threitol, 5 μl 5% NP40 and 22.5 μl molecular biology grade water [*Kurimoto et al., 2007*]). For experiments to determine the number of HIV DNA copies to measure $r$ and $q$, the DRAQ7-negative subset was sorted. Fixed, p24-positive, eFluor660-negative lymph node cells were single-cell sorted two days post-infection into 96-well plates containing 5 μl of PKD buffer (Qiagen, Hilden, Germany) with 1:16 proteinase K solution (Qiagen) (*Thomsen et al., 2016*). Sorted plates were snap frozen and kept at −80°C until ready for PCR. For analysis by flow cytometry, a minimum of 50,000 cells were collected per data point.

## Time-lapse microscopy and image analysis

For imaging infection by time-lapse microscopy, cell density was reduced to $5 \times 10^4$ cells/ml and cells were attached to ploy-l-lysine (Sigma-Aldrich) coated optical six-well plates (MatTek, **Ashland, MA**). Infections with and without EFV were imaged in tandem using a Metamorph-controlled Nikon TiE motorized microscope with a Yokogawa spinning disk with a 20x, 0.75 NA phase objective in a biosafety level three facility. Excitation sources were 488 (GFP) and 561 (mCherry) laser lines and emission was detected through a Semrock Brightline quad band 440–40/521–21/607-34/700-45 nm filter. Images were captured using an 888 EMCCD camera (Andor, Belfast, UK). Temperature (37°C), humidity and $CO_2$ (5%) were controlled using an environmental chamber (OKO Labs, Naples, Italy). Fields of view were captured every 20 min. To facilitate automated image analysis of time-lapse experiment data, mCherry expressing G2 clone cells were used as targets and E7 clone cells used as infected donors. The number of live cells was measured as the number of cells expressing mCherry since intracellular mCherry protein is soluble and hence lost upon cell death when cellular membrane integrity is compromised. The number of live infected cells was measured as the number of cells expressing both mCherry and GFP. Three independent experiments were performed. Movies were analyzed using custom code developed with the Matlab R2014a Image Analysis Toolbox. Images in the mCherry channel were thresholded and the imfindcircle function used to detect round objects within the cell radius range. Cell centers were found. GFP signal underwent the same binary thresh-olding. The number of mCherry-positive 16 pixel$^2$ squares around the cell centers was used as the as

the number of total target cells at each time-point, and the number of squares double positive for fluorescence in the GFP channel was used as the number of infected target cells.

## Determination of HIV DNA copy number in individual cells

96-well plates of cells previously sorted at 1 cell per well were thawed at room temperature and spun down. Fixed cells were de-crosslinked by incubating in a thermocycler at 56°C for 1 hr. The lysate from each well was split equally over 10 wells (2.5 μl each well after correction for evaporation) for E7 RevCEM or four wells (6.8 μl each well after correction for evaporation) for lymph nodes, containing 50 μl of Phusion hot start II DNA polymerase (New England Biolabs, Ipswich, MA) PCR reaction mix (10 μl 5X Phusion HF buffer, 1 μl dNTPs, 2.5 μl of the forward primer, 2.5 μl of the reverse primer, 0.5 μl Phusion hot start II DNA polymerase, 2.5 μl of DMSO and molecular biology grade water to 50 μl reaction volume). Two rounds of PCR were performed. The first round reaction amplified a 700 bp region of the reverse transcriptase gene using the forward primer 5' CCTACACCTG TCAACATAATTGGAAG 3' and reverse primer 5' GAATGGAGGTTCTTTCTGATG 3'. Cycling program was 98°C for 30 s, then 34 cycles of 98°C for 10 s, 63°C for 30 s and 72°C for 15 s with a final extension of 72°C for 5 min. 1 μl of the first round product was then transferred into a PCR mix as above, with nested second round primers (forward 5' TAAAAGCATTAGTAGAAATTTGTACAGA 3', reverse 5' GGTAAATCCCCACCTCAACAGATG 3'). The second round PCR amplified a 550 bp product which was then visualized on a 1% agarose gel. PCR reactions were found to work best if sorted plates were thawed no more than once, and plates which underwent repeated freeze-thaw cycles showed poor amplification.

## Correction of raw number of detected DNA copies for detection sensitivity

A stochastic simulation in Matlab was used to generate a distribution for the number of positive wells per cell for each mean number of DNA copies per cell $\lambda$. The probability for a DNA copy to be present within a given well and be detected was set as $\sigma/w$, where $\sigma$ was the detection sensitivity calculated as the number of ACH-2 with detectable integrations divided by the total number of ACH-2 cells assayed (38/166, $\sigma = 0.23$), and $w$ was the number of wells. A random number $m$ representing DNA copies per cell from a Poisson distribution with a mean $\lambda$ was drawn, and a vector $R$ of $m$ random numbers from a uniform distribution was generated. If there existed an element $R_i$ of the vector with a value between 0 and $\sigma/w$, the first well was occupied. If an element existed with a value between $\sigma/w+\gamma$ and $2(\sigma/w)$, where $\gamma <<1$, the second well was occupied, and if between $(\sigma/w +\gamma)(n-1)$ and $n(\sigma/w)$, the $n$th well was occupied. The sum of wells occupied at least once was determined, and the process repeated $j$ times for each $\lambda$, where $j$ was the number of cells in the experimental data. A least squares fit was performed to select $\lambda$ which best fit the experimental results across well frequencies, and mean and standard deviation for $\lambda$ was derived by repeating the simulation 10 times.

## Fit of the EFV response for single infections using IC$_{50}$ and Hill coefficient

To obtain $d$, we normalized *Equation (2)* by the fraction of infected cells in the absence of drug (*Sigal et al., 2011*) to obtain Tx = (infected targets with EFV)/ (infected targets no EFV)= $((1- (1\ r)^{\lambda/d})$ $(1-q)^{\lambda/d})/ ((1- (1\ r)^{\lambda}) (1-q)^{\lambda})$. We approximate the result at small $r$, $q$ to Tx = $(1- e^{-r\lambda/d})\ e^{-q\lambda/d}/ (1- e^{-r\lambda})\ e^{-q\lambda}=eq^{q\lambda(1-1/d)} ((1- e^{-r\lambda/d}) / (1- e^{-r\lambda}))$. Expanding the exponentials we obtain Tx = $(1 + q\lambda(1–1/d)) ((-r\lambda/d)/-r\lambda) = (1 + q\lambda(1–1/d))(1/d)$. We note that at $\lambda <1$, $q\lambda(1–1/d)<<1$, and hence Tx $\cong 1/d$. Tx was measured from the experiments to obtain $d$ values at the EFV concentrations used for cell-free infection, where $\lambda < 1$. To obtain a fit of $d$ as a function of the concentration of drug that gives half-maximal inhibition (IC$_{50}$) and Hill coefficient (h) for EFV, we used the relation for the fraction cells remaining infected in the face of drug (*Canini and Perelson, 2014*), whose definition is equivalent to Tx at $\lambda < 1$:

$$\frac{1}{d} = 1- = 1 - \frac{[EFV]^h}{[EFV]^h+IC_{50}^h}. \tag{4}$$

## Measurement of *r*

Cell-free supernatant used in infection was derived as follows: $10^6$ cells/ml were infected with $2 \times 10^8$ NL4-3 viral copies/ml (~20ng p24 equivalent) for 2 days. Thereafter 0.2% of the infected cells from the cell-free infection were added to $10^6$ cells/ml target cells. The infected supernatant from the coculture 2 days post-infection was filtered using a 0.45 µm filter (Corning) and added to cells at a 1:8 dilution, where the dilution was calibrated to result in non-saturating infection in terms of GFP expression. A fraction of the cells were sorted into lysis buffer at one cell per well 1 day post infection, split over four wells, and PCR performed as described above to determine HIV copy number per cell. The remaining cells from the same infection were used to determine frequency of DRAQ-7-negative, GFP-positive cells 2 days post-infection using flow cytometry.

## Measurement of *q*

Cell-free supernatant used in infection was derived as for *r*, except that 1 day before harvesting of the viral supernatant from infected cells, infected cells were washed twice with PBS and serum-free growth medium added. At the same time, the target cells for the infection were washed twice with PBS and serum-free growth medium added. Cells were split into two wells, and EFV to a final concentration of 40 nM was added to one of the wells. Supernatant was harvested and filtered as described for *r*, and added to cells at a 1:2 dilution. A fraction of the cells were sorted into lysis buffer at one cell per well 1 day post-infection, split over four wells, and PCR performed as described above to determine HIV copy number per cell. The remaining cells from the same infection were used to determine the frequency of live and dead cells two days post-infection. The concentration of live cells was measured using the TC20$^{TM}$ automated cell counter (Bio-Rad) with trypan blue staining (Lonza).

## Acknowledgements

This work was supported by National Institutes of Health Grant R21MH104220. AS was supported by a Human Frontiers Science Program Career Development Award CDA 00050/2013. RAN is supported by the European Research Council through grant Stg. 260686. LJ and JH are supported by a fellowship from the South African National Research Foundation. IMF is supported through a Sub-Saharan African Network for TB/HIV Research Excellence (SANTHE, a DELTAS Africa Initiative (grant #DEL-15–006)) fellowship, and a Poliomyelitis Research Foundation fellowship 17/59. Open access publication of this article has been made possible through support from the Victor Daitz Information Gateway, an initiative of the Victor Daitz Foundation and the University of KwaZulu-Natal.

## Additional information

### Competing interests

Richard A Neher: Reviewing editor, *eLife*. The other authors declare that no competing interests exist.

### Funding

| Funder | Grant reference number | Author |
| --- | --- | --- |
| Human Frontier Science Program | CDA 00050/2013 | Alex Sigal |
| European Research Council | Stg. 260686 | Richard A Neher |
| DELTAS Africa Initiative | Graduate Fellowship (DEL-15–006) | Isabella Markham Ferreira |
| National Research Foundation | Graduate Fellowship | Laurelle Jackson Jessica Hunter |
| National Institutes of Health | R21MH104220 | Alex Sigal |
| Poliomyelitis Research Foundation | Graduate Fellowship | Isabella Markham Ferreira |

The funders had no role in study design, data collection and interpretation, or the decision to submit the work for publication.

## Author contributions

Laurelle Jackson, Conceptualization, Data curation, Validation, Investigation, Visualization, Methodology, Writing—original draft, Project administration, Writing—review and editing; Jessica Hunter, Data curation, Validation, Investigation, Visualization, Methodology, Writing—original draft; Sandile Cele, Isabella Markham Ferreira, Andrew C Young, Yashica Ganga, Mikael Boulle, Investigation, Methodology; Farina Karim, Rajhmun Madansein, Resources, Project administration; Kaylesh J Dullabh, Resources, Data curation, Validation, Investigation; Chih-Yuan Chen, Resources, Investigation; Noel J Buckels, Khadija Khan, Resources; Gila Lustig, Resources, Investigation, Methodology, Project administration; Richard A Neher, Conceptualization, Methodology; Alex Sigal, Conceptualization, Resources, Data curation, Software, Formal analysis, Supervision, Funding acquisition, Validation, Investigation, Visualization, Writing—original draft, Project administration, Writing—review and editing

## Author ORCIDs

Andrew C Young (iD) http://orcid.org/0000-0003-3616-7956
Richard A Neher (iD) http://orcid.org/0000-0003-2525-1407
Alex Sigal (iD) http://orcid.org/0000-0001-8571-2004

## Ethics

Human subjects: Lymph nodes were obtained from the field of surgery of participants undergoing surgery for diagnostic purposes and/or complications of inflammatory lung disease. Informed consent was obtained from each participant, and the study protocol approved by the University of KwaZulu-Natal Institutional Review Board (approval BE024/09).

## Decision letter and Author response

Decision letter https://doi.org/10.7554/eLife.30134.039
Author response https://doi.org/10.7554/eLife.30134.040

# Additional files

## Supplementary files

• Supplementary file 1. S Table 1: Parameters and definitions.
DOI: https://doi.org/10.7554/eLife.30134.023

• Supplementary file 2. S Table 2: Measurement of $r$ and $q$
DOI: https://doi.org/10.7554/eLife.30134.024

• Supplementary file 3. S Table 3: Participant information.
DOI: https://doi.org/10.7554/eLife.30134.025

• Source code 1. script1r2.m: Matlab source code for *Figure 1*.
DOI: https://doi.org/10.7554/eLife.30134.026

• Source code 2. script2r2: Matlab source code for copy number correction for RevCEM clones based on HIV copy detection efficiency in ACH-2 cells.
DOI: https://doi.org/10.7554/eLife.30134.027

• Source code 3. script3r2.m: Matlab source code for cell-free and coculture infection model fits for RevCEM cells infected with wild-type HIV in *Figure 2*.
DOI: https://doi.org/10.7554/eLife.30134.028

• Source code 4. Script4.m: Matlab source code for histograms of raw HIV copy numbers for cell-free and coculture infection and fits of different distributions to coculture copy number frequencies.
DOI: https://doi.org/10.7554/eLife.30134.029

• Source code 5. Script5.m: Matlab source code for image analysis of time-lapse data for *Figure 2—figure supplement 5*.

DOI: https://doi.org/10.7554/eLife.30134.030

• Source code 6. script6r2.m: Matlab source code for cell-free and coculture infection model fits for RevCEM cells infected with L100I mutant HIV in *Figure 3*.
DOI: https://doi.org/10.7554/eLife.30134.031

• Source code 7. Script7.m: Matlab source code for copy number correction for lymph node cells based on HIV copy detection efficiency is ACH-2 cells.
DOI: https://doi.org/10.7554/eLife.30134.032

• Source code 8. script8r2.m: Matlab source code for cell-free and coculture infection model fits for lymph node cells infected with K103N mutant HIV in *Figure 6*.
DOI: https://doi.org/10.7554/eLife.30134.033

• Transparent reporting form
DOI: https://doi.org/10.7554/eLife.30134.034

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

## Appendix 1

DOI: https://doi.org/10.7554/eLife.30134.035

# Supplementary Mathematical Analysis for Jackson *et al.*

## I Derivation of infection model assuming independence of infection and cell death

We consider a model of infection where is the probability of one HIV DNA copy to successfully infect the cell and is the probability of the cell to die as a result of the infection attempt. The probability of being infected when virions enter the cell is:

$$P_n = (1 - (1 - r)^n)(1 - q)^n. \tag{AE1}$$

If the number of infection attempts, $n$, is Poisson distributed with mean $\lambda$, the fraction of cells that are productively infected is:

$$
\begin{aligned}
P_\lambda &= e^{-\lambda} \sum_n \frac{\lambda^n}{n!} P_n = e^{-\lambda} \sum_n \frac{\lambda^n}{n!}(1 - (1-r)^n)(1-q)^n \\
&= e^{-\lambda} \sum_n \frac{(\lambda(1-q))^n}{n!} - \frac{(\lambda(1-r)(1-q))^n}{n!} \\
&= e^{-\lambda q}(1 - e^{-\lambda r(1-q)}).
\end{aligned}
\tag{AE2}
$$

We introduce a drug strength value $d$, where $d = 1$ in the absence of drug and $d>1$ in the presence of drug. In the presence of drug, $\lambda$ is decreased to $\lambda/d$. The drug therefore tunes $\lambda$. The probability of a cell to be infected and live given drug strength $d$ is therefore:

$$P_{\lambda/d} = e^{-\lambda q/d}(1 - e^{-\lambda r(1-q)/d}). \tag{AE3}$$

Experimentally, we measure $P_\lambda$, the number of infected cells, and $L_\lambda$, the number of live cells remaining two days post-infection for the cell line.

Assuming a Poisson distribution of the number of infection attempts, the probability of being productively infected is

$$P_\lambda = 1 - e^{-\lambda r} \quad \Rightarrow \quad r = -\frac{\log(1 - P_\lambda)}{\lambda}. \tag{AE4}$$

Similarly, the probability of a cell to live is

$$L_\lambda = e^{-\lambda q} \quad \Rightarrow \quad q = -\frac{\log(L_\lambda)}{\lambda}. \tag{AE5}$$

## II Derivation of infection model assuming dependence of infection and cell death

Cellular infection and death may be the result of independent processes. For example, cell death may be a programmed response to DNA damage induced by the infection attempt and therefore may occur regardless of whether or not the cell is successfully infected. Infection and death may also be dependent. One example of dependence is the cytotoxicity mediated by expressed viral proteins after integration into the host genome.

In this case, the probability of productive infection and survival of the cell given $n$ attempts is

$$P_n = \sum_{m=1}^{n} \binom{n}{m}(r'(1-q'))^m (1-r')^{n-m} \tag{AE6}$$

Here, $r'$ and $q'$ are the probabilities for infection and death respectively. Their relationship to the experimentally measured $r$ and $q$ are discussed in the next section. $[r'(1 - q')]^m$ is the probability that $m$ infection attempts are successful and none of the attempts triggers cell

death, while $(1-r')^{n-m}$ accounts for $n-m$ unsuccessful infections. To obtain the total probability of productive infection without cell death given $n$ attempts, we sum over all cases with $m' \geq 1$ sucessful infections.

Realizing that:

$$\sum_{m=1}^{n} \binom{n}{m}(r'(1-q'))^m (1-r')^{n-m} = \sum_{m=0}^{n} \binom{n}{m}(r(1-q))^m (1-r')^{n-m} - (1-r')^n$$
$$= (r'(1-q') + (1-r'))^n - (1-r')^n, \qquad \text{(AE7)}$$

we can simplify the above to:

$$P_n = (r'(1-q') + (1-r'))^n - (1-r')^n = (1-r'q')^n - (1-r')^n. \qquad \text{(AE8)}$$

Averaging this over a Poisson distribution with mean $\lambda$, we find:

$$e^{-\lambda} \sum_n \frac{\lambda^n}{n!} P_n = e^{-\lambda} \sum_n \frac{\lambda^n}{n!}((1-r'q')^n - (1-r')^n) = e^{-\lambda r' q'} - e^{-\lambda r'}. \qquad \text{(AE9)}$$

## III Equivalence of the two models

The two models predict same dependence on $\lambda$ (the difference of two exponentials) but the parameters $r, q$ and $r', q'$ play slightly different roles. The models are the same if:

$$r'q' = q \quad \text{and} \quad r' = r(1-q) + q. \qquad \text{(AE10)}$$

Hence the two models are a re-parametrization of the same process. The experimental measurement of $r$ and $q$, as described in the main manuscript, consists of estimating $r$ as the fraction of infected cells when the fraction of dead cells is low due to a low number of measured DNA copies per cell. The death rate $q$ is derived by measuring the number of live cells remaining when the number of HIV DNA copies per cell is known. If the probability of death is dependent on infection, then death is only of infected cells and is a product of the probability to be infected and die ($r'q'$). Likewise, the underlying probability of infection $r'$ will account for the fraction dead cells ($q$) and any cells which are infected but not dead ($r(1-q)$).

