## [Decision Letter]

Thank you for submitting your article "Incomplete inhibition of HIV infection results in more HIV infected lymph node cells by reducing cell death" for consideration by *eLife*. Your article has been evaluated by Wenhui Li (Senior Editor) and three reviewers, one of whom is a member of our Board of Reviewing Editors. The reviewers have opted to remain anonymous.

The reviewers have discussed the reviews with one another and the Reviewing Editor has drafted this decision to help you prepare a revised submission.

Summary:

Jackson et al. mathematically predicted and experimentally demonstrated that in lymph node cells, intermediate antiviral (anti-HIV) drug concentrations can allow infected cells to live longer by reducing multiplicity of infection. Thus, intermediate drug concentration can presumably facilitate the persistence of HIV. The result is interesting and can have clinical implications.

Reviewer #1's concern is quite substantial. Other reviewers agreed to this concern. "I strongly believe that infection probability r and death probability q should be measured experimentally. This is to eliminate fitting parameters and to ensure quality of the model."

Comments from Reviewers 2 and 3 are attached.

*Reviewer #2:*

The studies presented by Sigal et al. examine the interrelationship between cell death and the number of DNA copies of HIV that are detected per individual infected cell. Previous studies from a number of groups, including Sigal's, have indicated that cell-to-cell infection leads to infected cells with a higher number of integrated proviruses. The single cell PCR data provided by Sigal et al. support a hypothesis that the cell-to-cell infection not only leads to higher copy number of proviruses, but also generally leads to more rapid cell death. Using drugs or antibodies at low levels that do not fully inhibit infection, they found that low levels of inhibition paradoxically can give rise to a larger number of surviving infected cells. They also appear to find lower rates of cell death under conditions of partial inhibition. The studies are generally well conceived, and the results generally support the conclusions. The overall quality of the data could be improved by showing additional controls and increasing the numbers of cells analyzed to enhance the confidence in the copy number estimates. On the cell death front, it would be helpful for their hypothesis to show a time course with the cell death depicted over time. Lastly, an important detail not mentioned, is whether the use of lymph node cells is critical or not for the phenotypes that they describe. The study may have implications for the establishment of viral reservoirs in the context of poorly controlled infection or infections with some degree of drug resistance.

1) The authors use the ACH2 cell line to test the ability of their assay to detect single copy integrations. Why do they only spread each cell over 4 wells in control studies as opposed to the 10 wells used in their experimental studies? The rationale for using different dilution schemes should be explained. A comparison of the efficiency of PCR at the different dilutions would also be informative.

The ACH2 cell line yields fewer than 1 copy per cell, which may be a limitation of the PCR assay, but also could be reflective of heterogeneity in a cell line that is assumed to be clonal and a uniform karyotype. It is not described how recently the ACH2 cell line that they are using has been cloned. Repeating the ACH2 studies with a subclone would be beneficial.

Figure 2 would be more informative if it showed the results of both cell-free virus infection versus coculture infection. The numbers of cells analyzed in Figure 2C (n=34) does not appear sufficient to provide a robust sense of the distribution of the DNAs in the infected cells. Is this a bimodal distribution? Histogram comparison of cell-free should also be used for comparison.

2) Figure 4.

What happens with partial antibody inhibition of cell-free infection? These studies should be performed with infection from cell-free virus. To further test their hypothesis partial inhibition of cell-free infection should only decrease infected cells.

3) Figure 5.

As controls for cell death measurements, the viability of the uninfected control lymph node cells, treated and untreated, should also be illustrated. In the literature lymph node cells (mostly tonsilar) are very prone to cell death, and it is important to understand to what extent they are measuring virus-induced cell death. In addition, a time course of the cell death observed in the infections may would also be helpful to evaluate their hypothesis that the increase of infected cells in the partially treated cells is due to increased infected cell survival – i.e. decreased death.

4) Are the phenotypes described in the primary lymph node cells also observed for peripheral blood lymphocytes? Or are there differences between the peripheral blood versus the lymph node T cells.

*Reviewer #3:*

This is an interesting and worthwhile paper that examines the effects of incomplete inhibition of HIV infection with reverse transcription inhibitors. The experimental results presented are in accordance with model predictions. However, the presentation of the model should be improved as detailed below. Also, I found the labeling and numbering of the supplemental figures confusing. I have no substantial concerns and this the paper should be published after minor revision.

1) The terms and concepts in the paper are not clearly defined. In the first paragraph of the Results, you mention each donor to target transmission. It is not clear what a transmission is. Does it refer to viral entry of either a free virus or a virus (or genome) by cell-to-cell transmission? It needs to be defined. Second define what you mean by infection. Does a cell have to produce virus to be considered infected? Does the virus have to integrate or is it sufficient to simply reverse transcribe? Is a latently infected cell an infected cell in your model? Also, when you say drug therapy increases the number of live infected cells do you mean live productively infected cells, live HIV DNA^+^ cells, etc.

2) The assumption that all transmissions have equal probabilities to infect target cells seems to ignore the possibility that some virions carry defective genomes while others do not. This does not seem realistic. In your experimental system is the ratio of HIV RNA to TCID50 close to one so you can ignore defective particles?

3) You assume productive infection and death are independent events. While the events may be independent the probability of death is certainly not independent of whether a cell is productively infected or not. Further the probability of death is time dependent. The probability a cell dies one hour after viral entry (infection?) is clearly quite different than the probability it dies days after infection. You may want to define q as the probability a cell has died by time t after infection and the same for Pλλ where t is 2 days or 4 days for your various experiments.

4) Results, second paragraph. You claim antiretroviral drugs reduce the number of infecting virions. This implies that by infecting you must mean the virus reverse transcribes. In standard viral dynamic models the effects of ART are to reduce the infection probability, i.e., r in your model not λ. Thus it is important to clarify your definitions.

5) You introduce the drug effect as a constant d in the second paragraph of the Results. Later in the paper you make reference to IC_50_ and Hill coefficients for the drug. These need to be tied together in an explicit manner. In viral dynamic modeling the effectiveness of a drug, epsilon, (eps) is introduced where eps=1 is a 100% effective drug, e.g. stops all reverse transcription, and where eps=0 means the drug has no effect. Then a pharmacodynamic model is used to relate eps with the drug concentration, C, e.g. eps= C^n/(IC50^n + C^n). Thus rather than λ/d you could model the drug effect by (1-eps)λ and use the above formula to link the effect to drug concentration. See for example, Canini and Perelson, 2014. Equation 3 in your manuscript accomplished the same, but the above seems more straightforward.

6) Results, fourth paragraph. When you measure the number of reverse transcribed copies of viral DNA I assume you are measuring both integrated and unintegrated DNA – please state this explicitly as later you seem to imply you are measuring integrated DNA, e.g. Results, sixth paragraph.

7) The current work focuses only on reverse transcriptase inhibitors. Model therapy uses combination therapy including integrase inhibitors and protease inhibitors. Expanding the discussion of the clinical implications of this work to include combination therapy would be worthwhile.

[Editors' note: further revisions were requested prior to acceptance, as described below.]

Thank you for submitting your article "Incomplete inhibition of HIV infection results in more HIV infected lymph node cells by reducing cell death" for consideration by *eLife*. Your article has been favorably evaluated by Wenhui Li (Senior Editor) and three reviewers, one of whom, Wenying Shou (Reviewer #1), is a member of our Board of Reviewing Editors. The following individual involved in review of your submission has agreed to reveal his identity: Alan Perelson (Reviewer #3).

The reviewers have discussed the reviews with one another and the Reviewing Editor has drafted this decision to help you prepare a revised submission.

We want you to address the following comments. Note that we are not asking you to do further experiments. For the sake of record-keeping, we are enclosing all comments.

*Reviewer #1:*

I am satisfied with their measurements.

In general, I suggest breaking up long sentences (especially the one in Abstract, also the end of the third paragraph in the Introduction) for readability. Sentences that last more than three lines are generally more difficult to read than shorter sentences.

*Reviewer #2:*

The authors have very rigorously addressed the critiques by performing new experiments and modifying the text. The response to the reviews is particularly comprehensive and thorough. The resulting new data are supportive of the original model and hypothesis. The authors should be commended.

*Reviewer #3:*

In this revised version on the paper the authors have clarified their definitions by incorporating a supplemental table and additional text describing their model. However, this clarification has raised the following issues that need to be resolved:

1) As defined in the Results, λ is the number of infection attempts, where one attempt is measured as one HIV DNA copy. Thus, λ is an integer. The formula, Equation 1, makes sense when λ is an integer. However, what is determined experimentally is not the integer number of DNA copies but rather the mean number of copies per cell, which is not necessarily an integer. To make sense Equation 1 should be reformulated as a conditional probability of a cell being infected and alive given x attempts, call this P(I | x). What I believe the authors want is the probability of a cell being infected and alive, which is then P(I) =Sum P(I |x) (p(x)), where p(x) is the probability of having x infection attempts, and the sum is over all non-negative values of x. One could then assume, for example, that the number of attempts is Poisson distributed with mean λ. Then the resulting formula for P(I) would involve the mean value of the number of attempts. A better choice would be the empirical distribution that the authors measured. Note that what is actually measured is the number of infected cells that are alive at a certain time, i.e. P(I), and the mean number of attempts, what I called λ above.

2) According to the model q should be independent of the number of attempts. Can this be tested by using different drug concentrations, which would vary the mean number of attempts? If you already have experimental data on this, then include them. Otherwise, just discuss it.

3) In the original manuscript where Equation 1 was derived the authors assumed q was independent of r. In the second paragraph of the Results the authors say q is the probability of a cell to die post reverse transcription, which seems more sensible to me. Under the original assumption of q and r being independent, a cell in which the attempt was unsuccessful would have the same probability of dying as a cell in which reverse transcription succeeded. With q now defined as in the second paragraph of the Results, the fundamental formula given by Equation 1 may not be correct and deserves a more thorough discussion. I think the fundamental process being described is that at each attempt a cell is either infected with probability r, or not infected with probability 1-r. If it is infected, i.e. had a successful reverse transcription, then with probability q it remains alive and with probability 1-q it dies. Assume there are x attempts. Let z be the number of cells infected after x attempts. Then z is binomially distributed, i.e. z= Bin (x,r). Further, if we are interested in the number of cells infected and alive after x attempts, then P(I|x) = Sum_z=1 to x (x choose r) r^z (1-r)^(x-z)(1-q)^z, where the factor (1-q)^z is the probability of a cell surviving after each of the z infections. Simplifying, P(I|x) = [r(1-q) + (1-r)]^x, which is not the same as Equation 1. The number of attempts, x, is again random and as above one can convert this conditional probability into P(I) by assuming a Poisson (or some other distribution such as the empirically measured one) for the number of attempts p(x).

4) In Supplementary file 1 the authors say the way they measure the probability a cell is alive after λ attempts is by computing the concentration of cells alive with λ attempts divided by the concentration of cells alive with no transmissions. However, since the concentration of cells alive with no transmissions can approach zero as time goes on it is clear that this fraction need not be less than 1. Also, as noted above the authors need to use conditional probabilities and derive formulas that involve the mean number of attempts. I would suggest they compute the ratio of the probability a cell is infected to the probability a cell in not infected and alive, which is what they measure, and see if they can derive a formula for q.

---

## [Author Response]

Reviewer #1's concern is quite substantial. Other reviewers agreed to this concern. "I strongly believe that infection probability r and death probability q should be measured experimentally. This is to eliminate fitting parameters and to ensure quality of the model."

In order to address this important point, we designed two sets of experiments, one set for measuring r and the other for q. The definition for r and q is now sharpened as suggested by reviewer 3 in Supplementary file 1 (as described in the response to reviewer 3 comments).

Our general approach was to use cell-free infection to avoid the high multiplicity of infection and broad distribution of HIV copy number observed in cell-to-cell spread. We then measured the fraction of live infected cells (P_λ_) or cell viability (L_λ_) post-infection. For each set of experiments we determined the number of infection attempts λ, which was measured as the number of total HIV DNA copies per cell, both integrated and unintegrated. This enabled us to calculate p and q (see below). For q, we compared infected cells to cells infected but treated with 40nM efavirenz, sufficient to reduce reverse transcription to negligible levels in cell-free infection (see Figure 2D). Hence we measured the effect on cell viability of infection from the reverse transcription step onward, as this is what is compared in all our experiments.

Two independent experiments were performed in the cell line per parameter, with each experiment including cell sorting followed by PCR to obtain λ. The specific approach for each parameter was as follows:

Measurement of r (probability of infection per transmission)

1) r was measured using cell-free infection followed by flow cytometry 2 days post-infection to determine frequency of live infected cells P_λ_, as measured by the fraction of live GFP positive cells normalized by the maximum fraction of cells which can become GFP positive upon infection in our system (70%).

2) λ, the number of viral transmissions per cell, was defined as the average number of detected HIV DNA copies per cell both integrated and unintegrated (see response to reviewer 3 below), as detected by splitting a single cell over multiple wells and performing PCR for HIV DNA.

3) Given that live cells were selected, r = 1-(1-P_λ_)^1/λ^

Measurement of q (probability of cell death per transmission)

1) The fraction of live cells remaining post-reverse transcription, defined as L_λ_ =[live cells with infection]/[live cells when reverse transcription blocked], was measured with cell-free infection by determining the concentration of live cells using cell counting with trypan blue exclusion. To prevent cell numbers changing due to cell division, proliferation was blocked by serum starvation.

2) λ was defined and measured as with r.

3) q = 1-L_λ_^1/λ^.

Results of experiments are shown in the new Figure 2—figure supplement 4.

The results are also summarized in Supplementary file 2.

We used the measured values to calculate P_λ_ using Equation 2. The results are presented in Figure 2E, green line.

The measured q value is higher than our fit (0.14 versus 0.08) and the r value lower (0.24 versus 0.40). However, the pattern of the peak is not strongly affected. What this demonstrates is that the measured r and q are in the correct range in the cell line system to explain the observed peak.

This result is now presented in the main text in the Results section:

“In order to determine whether the fitted *r* and *q* values were within a reasonable range, we measured these values experimentally. […] The solution to Equation 2 using these values was very similar to the solution with the fitted values, indicating that the fitted values gave a reasonable approximation of the behavior of the system (Figure 2E, green dotted line).”

The methods to measure r and q are presented in the Materials and methods section:

“Measurement of *r*

Cell-free supernatant used in infection was derived as follows: 10^6^ cells/ml were infected with 2x10^8^ NL4-3 viral copies/ml (~20ng p24 equivalent) for 2 days. […] The remaining cells from the same infection were used to determine frequency of DRAQ-7 negative, GFP positive cells 2 days post-infection using flow cytometry.”

“Measurement of *q*

Cell-free supernatant used in infection was derived as for *r*, except that one day before harvesting of the viral supernatant from infected cells, infected cells were washed twice with PBS and serum-free growth medium added. […] The concentration of live cells was measured using the TC20^TM^ automated cell counter (Bio-Rad) with trypan blue staining (Lonza).”

To measure r and q in lymph node cells is not practical. This is because the infection is of a subset of lymph node cells (~10%). Therefore, measurement of λ where most cells are uninfected would require months of running gels. Furthermore, it is very unlikely that we could measure q using the approach above, since a given frequency of cell death in the infectable cell subpopulation would be diluted to a much lower frequency over the whole cell population, and unlikely to be detected above the high background lymph node cell death.

Comments from Reviewers 2 and 3 are attached.Reviewer #2:[…] The overall quality of the data could be improved by showing additional controls and increasing the numbers of cells analyzed to enhance the confidence in the copy number estimates.

Additional controls were added as detailed in the replies to the reviewer’s specific comments. The number of cells analyzed increased as follows:

For ACH-2 cells, in total an additional 95 cells were analyzed, bringing to total analyzed cells to 166. This is now shown in Figure 2—figure supplement 1.

For the cell line infected with cell-free virus, an additional 12 cells were analyzed, bringing the total analyzed cells to 30. For coculture infection, an additional 26 cells were added, bringing the total analyzed cells to 60. These results are presented in Figure 2C with cell-free data now included as an inset.

With the new results, the mean number of corrected HIV DNA copies per cell was 17.

On the cell death front, it would be helpful for their hypothesis to show a time course with the cell death depicted over time.

We now include time-lapse microscopy data showing the depletion of target cells in the absence and presence of efavirenz. The experiments show the total number of live and live infected cells in the optical dish with no drug, and in the presence of efavirenz. These experiments were done before we finalized experimental conditions, so the input of wild type virus infected donor cells (20%) and the corresponding efavirenz concentrations (30nM) were higher. However, the data clearly shows that extensive death due to infection occurs in the absence of drug, and is prevented by efavirenz. The data is presented as Figure 2—figure supplement 5.

This is now described in the Results section:

“In order to investigate the dynamics of cell depletion due to cell-to-cell HIV spread and its modulation by addition of inhibitor, we performed time-lapse microscopy over a two day infection window. […] The deterioration in live cell numbers was averted by the addition of EFV (Figure 2—figure supplement 5).”

The method used is described in the Materials and methods:

“Time-lapse microscopy and image analysis

For imaging infection by time-lapse microscopy, cell density was reduced to 5x10^4^ cells/ml and cells were attached to ploy-l-lysine (Σ-Aldrich) coated optical six well plates (MatTek). […] The number of mCherry positive 16 pixel^2^ squares around the cell centers was used as the as the number of total target cells at each time-point, and the number of squares double positive for fluorescence in the GFP channel was used as the number of infected target cells.”

Lastly, an important detail not mentioned, is whether the use of lymph node cells is critical or not for the phenotypes that they describe.

In this study we concentrated on lymph nodes, which were reported to be sites of lower drug penetration as well as efficient HIV replication. However, we did try PBMCs, and observed no obvious fitness optimum, as reviewer 2 suggests may be a possibility since PBMCs are less prone to cell death relative to lymph node cells. The first set of 3 independent experiments using blood from one healthy donor showed a small peak at a very low EFV concentration. We then focused on the lower EFV concentration range but could not reproduce the peak in other donors. These results are presented as Figure 5—figure supplement 1.

It is described in the Results section:

“Peaks in the number of live infected cells in the face of drug may be specific to lymph node derived cells. Cell-to-cell infection of peripheral blood mononuclear cells (PBMC) with wild type HIV showed a slight peak at a very low EFV concentration in cells from one blood donor, which was not repeated in cells from two other donors (Figure 5—figure supplement 1).”

We add this observation to the Discussion:

“Physiologically, an infection optimum in the face of an antiretroviral drug may be important in lymph node cell HIV infection and may be less pronounced in infectable cells from peripheral blood.”

The study may have implications for the establishment of viral reservoirs in the context of poorly controlled infection or infections with some degree of drug resistance.

We thank the reviewer for this concise summary and add it to the Discussion:

“This study may have implications for the establishment of viral reservoirs in the context of poorly controlled infections, infections with some degree of drug resistance, or infections where some replication may take place in the face of ART, since infected cell survival is a pre-requisite for long term persistence.”

1) The authors use the ACH2 cell line to test the ability of their assay to detect single copy integrations. Why do they only spread each cell over 4 wells in control studies as opposed to the 10 wells used in their experimental studies? The rationale for using different dilution schemes should be explained. A comparison of the efficiency of PCR at the different dilutions would also be informative.

There is no reason for the difference except that we started with 10 well spreads in cell line coculture infections and realized that we can use less reagents to obtain the same data. We now state this in the Results section:

“Using fewer wells saved reagents without changing sensitivity, as demonstrated in the ACH-2 cell line (Figure 2—figure supplement 1C).”

We compared efficiency of HIV copy detection in ACH-2 cells between 4 and 10 well spreads and did not find a substantial difference, as the frequency in the 10-well split was 0.27 while the parental was 0.24. The data is presented in Figure 2—figure supplement 1,

We describe this in the Results section:

“Similar results were obtained when the ACH-2 cell line was subcloned or split over 10 wells (Figure 2—figure supplement 1C).”

The ACH2 cell line yields fewer than 1 copy per cell, which may be a limitation of the PCR assay, but also could be reflective of heterogeneity in a cell line that is assumed to be clonal and a uniform karyotype. It is not described how recently the ACH2 cell line that they are using has been cloned. Repeating the ACH2 studies with a subclone would be beneficial.

We subcloned the ACH-2 cells as described in the Materials and methods:

“Subcloning of ACH-2 cells

Cells from the parental ACH-2 cell line were diluted to 10 cells/ml in conditioned medium, with conditioned medium generated by culturing ACH-2 cells to 10^6^ cells/ml, then filtering through a 0.22µm filter (Corning). […] Clones were detected in 5% of wells and two clones, designated D6 and C3, were randomly chosen and further expanded.”

We checked two subclones, with 56 cells measured in the first subclone and 24 in the second. Frequency of detection in each was 0.21, which we judged was not substantial relative to the parental line. We describe the results in Figure 2—figure supplement 1C above.

Figure 2 would be more informative if it showed the results of both cell-free virus infection versus coculture infection. The numbers of cells analyzed in Figure 2C (n=34) does not appear sufficient to provide a robust sense of the distribution of the DNAs in the infected cells. Is this a bimodal distribution? Histogram comparison of cell-free should also be used for comparison.

We now include the cell-free distribution as an inset in Figure 2C. Frequency of HIV DNA copies was 0.23, as described in the response to general comment 3. We describe this in the Results section:

“We also detected the HIV copy number in 30 GFP positive cells infected by cell-free HIV. In this case, we detected either zero or one HIV copy per cell (Figure 2C inset). The frequency of single HIV DNA copies was 0.23, identical to the measured result in the ACH2 cell line.”

We checked the fit of the distribution resulting from coculture infection, which we expanded to 60 analyzed cells. While a distribution with two means fit the data better, we could not rule out that this was due to added fit parameters since the fitted distribution itself did not show peaks. Most likely, this means that the peak at 5 raw copies and no cells at 6 copies is a chance event that gives an impression of bimodality. The results are presented in Figure 2—figure supplement 2.

These results are described in the Results section:

“We assayed 60 cells and obtained a wide distribution of viral DNA copies per cell, which ranged from 0 to 9 copies (Figure 2C). […] The frequency of single HIV DNA copies was 0.23, identical to the measured result in the ACH-2 cell line.”

2) Figure 4.What happens with partial antibody inhibition of cell-free infection? These studies should be performed with infection from cell-free virus. To further test their hypothesis partial inhibition of cell-free infection should only decrease infected cells.

We now include the results for antibody inhibition of cell-free infection as Figure 4—figure supplement 1. As can be seen, antibody only decreases infected cells in cell-free infection.

This is described in the Results section:

“In contrast, cell-free infection in the face of b12 showed a sharp and monotonic drop in live infected cells for both wild type and mutant virus (Figure 4—figure supplement 1).”

3) Figure 5.As controls for cell death measurements, the viability of the uninfected control lymph node cells, treated and untreated, should also be illustrated. In the literature lymph node cells (mostly tonsilar) are very prone to cell death, and it is important to understand to what extent they are measuring virus-induced cell death.

We measured cell death in uninfected lymph node cells, untreated and treated with efavirenz. We did not find that efavirenz alone results in a peak of live cells (Figure 6—figure supplement 3).

This is discussed in the Results section:

“To examine if the observed peak in live cells may be due to EFV alone, we measured cell viability in lymph node cells from one of the study participants used in the above experiment as a function of EFV without infection. No dependence on EFV in the absence of infection was detected (Figure 6—figure supplement 3).”

In addition, a time course of the cell death observed in the infections may would also be helpful to evaluate their hypothesis that the increase of infected cells in the partially treated cells is due to increased infected cell survival – i.e. decreased death.

As discussed in the response to the reviewer’s general comment 3, we now include timelapse measurements of infection treated and untreated with the efavirenz which supports our hypothesis that the increase of infected cells in the partially treated cells is due to increased infected cell survival.

4) Are the phenotypes described in the primary lymph node cells also observed for peripheral blood lymphocytes? Or are there differences between the peripheral blood versus the lymph node T cells.

As discussed in the response to the reviewer’s general comment 4, the work we did perform on PBMCs showed little increase in infection with partial inhibition, consistent with the reviewer’s intuition that this may be a lymph node specific effect.

Reviewer #3:This is an interesting and worthwhile paper that examines the effects of incomplete inhibition of HIV infection with reverse transcription inhibitors. The experimental results presented are in accordance with model predictions. However, the presentation of the model should be improved as detailed below. Also, I found the labeling and numbering of the supplemental figures confusing. I have no substantial concerns and this the paper should be published after minor revision.

We thank the reviewer for the support. We have clarified the model parameters and how they are measured as detailed in the responses to the Reviewer’s specific comments below to address this very important point. We have ensured that all supplemental figures are labelled in *eLife* format and sequentially referenced in the text.

1) The terms and concepts in the paper are not clearly defined. In the first paragraph of the Results, you mention each donor to target transmission. It is not clear what a transmission is. Does it refer to viral entry of either a free virus or a virus (or genome) by cell-to-cell transmission? It needs to be defined. Second define what you mean by infection. Does a cell have to produce virus to be considered infected? Does the virus have to integrate or is it sufficient to simply reverse transcribe? Is a latently infected cell an infected cell in your model? Also, when you say drug therapy increases the number of live infected cells do you mean live productively infected cells, live HIV DNA^+^ cells, etc.

The reviewer has identified an important gap in our presentation of the results which we have now corrected. We now present a table of definitions for each parameter in our system and how it is defined and measured. This now appears in the manuscript as Supplementary file 1.

We make extensive use of the sharpened definitions, including in the Results section:

“We introduce a model of infection where each donor to target transmission leads to an infection probability *r* and death probability *q* per infection attempt, where each model parameter is defined in Supplementary file […] We define *L_λ_* as the probability of a cell to survive infection given λ infection attempts. Assuming infection attempts act independently, *L_λ_*=*(1-q)^λ^*.”

2) The assumption that all transmissions have equal probabilities to infect target cells seems to ignore the possibility that some virions carry defective genomes while others do not. This does not seem realistic. In your experimental system is the ratio of HIV RNA to TCID50 close to one so you can ignore defective particles?

Given that transmission is defined as the mean number of HIV DNA copies per cell and that this is what we measure, any virion which does not successfully reverse transcribe will not be detected. Therefore, the measurement of the number of transmissions per cell λ excludes virions defective in virus cycle stages leading up to an including reverse transcription. The fraction of defective viruses post-reverse transcription is captured in the term r, as HIV DNA copies with mutations which make them defective post-reverse transcription would not be able to produce viral proteins, our measure of successful infection. This is now described in the Results section:

“*r* and *q* capture the probabilities for a cell to be infected or die post-reverse transcription. For example, mutations which reduce viral fitness by decreasing the probability of HIV to integrate would reduce *r*, while mutations which reduce the probability of successful reverse transcription would reduce *λ*.”

3) You assume productive infection and death are independent events. While the events may be independent the probability of death is certainly not independent of whether a cell is productively infected or not. Further the probability of death is time dependent. The probability a cell dies one hour after viral entry (infection?) is clearly quite different than the probability it dies days after infection. You may want to define q as the probability a cell has died by time t after infection and the same for Pλ where t is 2 days or 4 days for your various experiments.

We thank the reviewer for bringing up the important point that cell death is a time-dependent process and the time-point at which it is quantified needs to be defined. We therefore corrected our definitions to state that the fraction of live infected cells is measured at 2 days post-infection for the cell lines and 4 days post-infection for lymph node cells, as the reviewer suggested. This now appears in Supplementary file 1 as discussed above.

4) Results, second paragraph. You claim antiretroviral drugs reduce the number of infecting virions. This implies that by infecting you must mean the virus reverse transcribes. In standard viral dynamic models the effects of ART are to reduce the infection probability, i.e., r in your model not λ. Thus it is important to clarify your definitions.

The reviewer is correct that you would expect the effect of the reverse transcriptase inhibitor to be included in r. However, our measurable transmission λ is a reverse transcribed copy of HIV DNA. Hence in our model r accounts for the probability of a reverse transcribed viral genome successfully integrating into the host genome and producing viral proteins, which are independent of the action of efavirenz. The way we set up our model, the action of the reverse transcriptase inhibitor is on λ, not r. We address this as described in the response to point 2 of the reviewer’s comments.

5) You introduce the drug effect as a constant d in the second paragraph of the Results. Later in the paper you make reference to IC50 and Hill coefficients for the drug. These need to be tied together in an explicit manner. In viral dynamic modeling the effectiveness of a drug, epsilon, (eps) is introduced where eps=1 is a 100% effective drug, e.g. stops all reverse transcription, and where eps=0 means the drug has no effect. Then a pharmacodynamic model is used to relate eps with the drug concentration, C, e.g. eps= C^n/(IC50^n + C^n). Thus rather than λ/d you could model the drug effect by (1-eps)λ and use the above formula to link the effect to drug concentration. See for example, Canini and Perelson, 2014. Equation 3 in your manuscript accomplished the same, but the above seems more straightforward.

The reviewer’s notation is more elegant than what we use since it directly introduces the effect of drug per virus into the equation. The reason we would argue to keep our current notation is that d is on the same scale as λ, which we think makes a ratio such as d/λ more intuitive. Thus d=λ would reduce the infection to single HIV DNA copies per cell. We have stated the conversion between d and ε in the Materials and methods section:

“we used the relation for the fraction cells remaining infected in the face of drug (Canini and Perelson, 2014), whose definition is equivalent to *Tx* at *λ<1*:

1d=1-=1-EFVhEFVh+IC50h (3).”

We also discuss the conversion in the Results section:

“This is equivalent to 1-ε in a commonly used model describing the effect of inhibitors on infection, where ε is drug effectiveness, with the 50% inhibitory drug concentration (IC_50_) and the Hill coefficient for drug action as parameter values (Canini and Perelson, 2014; Shen et al., 2008).”

6) Results, fourth paragraph. When you measure the number of reverse transcribed copies of viral DNA I assume you are measuring both integrated and unintegrated DNA – please state this explicitly as later you seem to imply you are measuring integrated DNA, e.g. Results, sixth paragraph.

We thank the reviewer for pointing out this omission and we now explicitly state that λ measures the number of HIV DNA copies, integrated and unintegrated. This is now reflected in Supplementary file 1 as described above, as well as in the Results section:

“In our experimental system, one infection attempt is measured as one HIV DNA copy, whether integrated or unintegrated.”

7) The current work focuses only on reverse transcriptase inhibitors. Model therapy uses combination therapy including integrase inhibitors and protease inhibitors. Expanding the discussion of the clinical implications of this work to include combination therapy would be worthwhile.

We agree with the reviewer that our analysis can be applied beyond reverse transcriptase inhibitors. We used efavirenz in our study due it being a common principal component of current first line antiretroviral therapy, with common drug resistance mutations. However, the infection optimum we describe should occur with other classes of antiretroviral drugs, since they all should decrease the multiplicity of infection between cells. We now include this point in the Discussion:

“We used EFV in our study since it is a common component of first line antiretroviral therapy, with frequent drug resistance mutations. […]The more complex outcome of partial inhibition of infection should also be considered in other infections where multiple pathogens infect one cell and host cell death is a possible outcome (Mahamed et al., 2017).”

[Editors' note: further revisions were requested prior to acceptance, as described below.]

We want you to address the following comments. Note that we are not asking you to do further experiments. For the sake of record-keeping, we are enclosing all comments.Reviewer #1:I am satisfied with their measurements.In general, I suggest breaking up long sentences (especially the one in Abstract, also the end of the third paragraph in the Introduction) for readability. Sentences that last more than three lines are generally more difficult to read than shorter sentences.

We have now broken up long sentences as follows.

Abstract:

“Using a computational approach, we found that partial inhibition of transmissions of multiple virions per cell could lead to increased numbers of live infected cells. If the number of viral DNA copies remains above one after inhibition, then eliminating the surplus viral copies reduces cell death.”

Introduction:

“Multiple infections per cell have been reported in cell-to-cell spread of HIV (Baxter et al., 2014; Boull et al., 2016; Dang et al., 2004; Del Portillo et al., 2011; Dixit Perelson, 2004; Duncan et al., 2013; Law et al., 2016; Reh et al., 2015; Russell et al., 2013; Sigal et al., 2011; Zhong et al., 2013). In this mode of HIV transmission, an interaction between the infected donor cell and the uninfected target results in directed transmission of large numbers of virions (Baxter et al., 2014; Groppelli et al., 2015; Hubner et al., 2009; Sowinski et al., 2008).”

Introduction:

“Both modes occur simultaneously when infected donor cells are cocultured with targets. However, the cell-to-cell route is thought to be the main cause of multiple infections per cell (Hubner et al., 2009).”

Introduction:

“However, more recent work investigating markers associated with HIV latency in the face of ART found that the average number of HIV DNA copies per cell is greater than one in 3 out of 12 individuals. This occurred in the face of ART in the CD3 positive, CD32a high CD4 T cell subset (Descours et al., 2017).”

Introduction:

“We observed that partially inhibiting infection with drug or antibody resulted in an increase in the number of live infected cells in both a cell line and in lymph node cells. This is, to our knowledge, the first experimental demonstration at the cellular level that attenuation of HIV infection can result in an increase in live infected cells under specific infection conditions.”

Results:

“To estimate this, we used PCR to detect the number of reverse transcribed copies of viral DNA in the cell by splitting each individual infected cell over multiple wells. We then detected the number of wells with HIV DNA by PCR amplification of the reverse transcriptase gene.”

Results:

“This is equivalent to 1-ε in a commonly used model describing the effect of inhibitors on infection. In this model, is drug effectiveness, with the 50 percent inhibitory drug concentration (IC_50_) and the Hill coefficient for drug action as parameter values (Canini Perelson, 2014; Shen et al., 2008).”

Discussion:

“If each HIV DNA copy increases the probability of cell death, reducing the number of HIV DNA copies without eliminating infection should lead to an increased probability of infected cell survival. This would consequently lead to an increase in the number of live infected cells.”

Reviewer #3:In this revised version on the paper the authors have clarified their definitions by incorporating a supplemental table and additional text describing their model. However, this clarification has raised the following issues that need to be resolved:1) As defined in the Results, λ is the number of infection attempts, where one attempt is measured as one HIV DNA copy. Thus, λ is an integer. The formula, Equation 1, makes sense when λ is an integer. However, what is determined experimentally is not the integer number of DNA copies but rather the mean number of copies per cell, which is not necessarily an integer. To make sense Equation 1 should be reformulated as a conditional probability of a cell being infected and alive given x attempts, call this P(I | x). What I believe the authors want is the probability of a cell being infected and alive, which is then P(I) =Sum P(I |x) (p(x)), where p(x) is the probability of having x infection attempts, and the sum is over all non-negative values of x. One could then assume, for example, that the number of attempts is Poisson distributed with mean λ. Then the resulting formula for P(I) would involve the mean value of the number of attempts. A better choice would be the empirical distribution that the authors measured. Note that what is actually measured is the number of infected cells that are alive at a certain time, i.e. P(I), and the mean number of attempts, what I called λ above.

We thank the reviewer for this comment. The reviewer is correct that our previous Equation 1 applies to infection attempts per cell, which are integer values. We have now reformulated the model to use the mean number of infection attempts, as described below and in the new Appendix 1: Supplementary Mathematical Analysis section. We used the Poisson distribution, the first suggestion of the reviewer, as we believe we need much more extensive experimental data to determine the exact empirical distribution.

We now consider the probability of a cell being infected and live given *n* HIV DNA copies, where n is an integer (Appendix 1-Equation 1).

If *P(n*) is Poisson with mean *λ*, the fraction of cells that are productively infected is as in Appendix 1-Equation 2.

We introduce a drug strength value *d*, where *d* = 1 in the absence of drug and *d >* 1 in the presence of drug. In the presence of drug, *λ* is decreased to *λ/d*. The drug therefore tunes *λ*. The probability of a cell to be infected and live given drug strength d is therefore as in Appendix 1-Equation 3.

Appdenxi 1-Equation 3 is now used to derive Figure 1, which is qualitatively similar to Figure 1 from the previous submission using Equation 2. The range of *q* values was modified to show the full range of infection behavior with the new model (Figure 1)

Experimentally, we measure *P_λ_*, the number of infected cells, and *L_λ_*, the number of live cells remaining two days post infection for the cell line.

Assuming a Poisson distribution of the number of infection attempts, the probability of being productively infected is as in Appendix 1-Equation 4.

Similarly, the probability of a cell to live is as in Appendix 1-Equation 5.

The derivation of the model above is included in the new Supplementary Mathematical Analysis section. The new definition of *P_λ_*and *L_λ_*is now updated in Supplementary file 1.

The experimentally determined values for *r* and *q* are now recalculated to be *r* = 0.28 and *q* = 0.15. This is now updated in Supplementary file 2.

Comparing this model to the model described by the previous model (Equation 2 in the previous submission), we note that the models give very similar results at *λ >* 1, but that our previous model underestimates the rate of decline in infection at *λ <* 1, as shown in Author response image 1 for L100I HIV mutant coculture infection of the cell line, where the effect was most pronounced.

To investigate whether the experimental results were above the fit line due to technical factors, we refined our experimental approach to exclude donor cell – target cell fusions, which may give a baseline in the infection. In the previous submission, we judged the fraction of fusions to be negligible and selected on target cells based on mCherry fluorescence. Here, we used the vital stain Cell Trace Far Red (CTFR) to label donor cells. Target cells still expressed mCherry. We therefore selected target cells based on mCherry, but excluded any target cells which were CTFR and mCherry double positive. Due to the extra fluorescence channel, it was necessary to change from our previous death detection dye with far-red emission (DRAQ7) to the blue spectrum emission death detection dye (DAPI). For this, we used a different flow cytometer (FACSAria Fusion) which had the 355nm laser for DAPI excitation. The gating strategy is now illustrated in Figure 2— Figure supplement 4 for wild type HIV infection.

It is described in the Results section: “To obtain the number of infected target cells, and specifically exclude donor cells or donor-target cell fusions, target cells were marked by the expression of mCherry. […] Donor-target cell fusions were excluded by excluding CTFR positive cells (see Figure 2— figure supplement 4 for gating strategy).” The results shown in Figure 2— figure supplement 4 are described in the Results section:

“While the percent of infected cells was reduced with drug, the concentration of live infected cells increased (Figure 2—figure supplement 4).”

We also describe the method in the Materials and methods section:

“To determine the number of live infected cells in reporter cell line experiments, E7 RevCEM reporter cells were infected as above used as donor cells. […] The coculture infection was pulsed with 100ng/ml DAPI (Σ) immediately before flow cytometry and the number of live infected targets cells was determined by the number of DAPI negative, CTFR negative and mCherry and GFP double positive cells on a FACSAria Fusion machine (BD Biosciences) using the 355, 488 and 633nm laser lines.”

We used the same approach for infection with the L100I HIV mutant, and the gating strategy is illustrated in the new Figure 3—figure supplement 2.

We describe the new figure in the Results:

“We next performed coculture infection (see Figure 3—figure supplement 2 for gating strategy).”

We also increased the number of data points for wild type and mutant infections in the cell line when we repeated the experiments. The number of data points increased from 5 to 7 for wild type HIV infection, and 6 to 8 for L100I mutant infection of the cell line (3 independent experiments each). For mutant infection, we concentrated on drug concentrations closer to the peak since that was the region with the most pronounced decline in infection. Excluding donor-target cell fusions using new gating strategy resulted in an incremental improvement to the data, as shown in Author response image 2 for wild type mutant HIV infection in the cell line:

**Author response image 2. respfig2:** 

These new results showed that the lower than expected decline in infection at high drug concentrations was not a technical artifact due to a baseline of infected donor cells which were fused to targets. This deviation from the model at high drug concentrations away from the peak infection with drug is now noted in the Results section:

“We note that both wild type and mutant coculture infection has data points above the fit line at the highest drug concentrations. This may be a limitation of our model at drug values much higher than observed at the infection optimum. In this range of drug values, our model predicts a more pronounced decline in the number of infected cells than is observed experimentally.”

We also include the observation in the Discussion section:

“Neither model accurately captures infection dynamics at high drug concentrations, away from the infection optimum. In this range, where the number of infection attempts per cell is much lower than 1, infection declined more slowly with drug than predicted.”

We feel this observation may be potentially important to understanding the HIV reservoir. However, the existence of a fitness optimum with drug is a separate issue, and our current model successfully describes how the fitness optimum is formed.

An important conclusion reached when we compared the fits before and after fusion exclusion, as well as the fit of the previous iteration of L100I mutant infection to the current experiments, is that small differences in the experimental data resulted in large differences in the value of *r*. Hence, *r* did not seem to be well constrained in the fits. As shown in Figure 1, *r*determines the amplitude of *P_λ_*. This is now emphasized in the Results section:

“Hence, the value of *r* strongly influences the amplitude of *P_λ_*.”

To better constrain *r* as a result of these new observations, we switched from normalizing the number of infected cells and *P_λ_*to the maximum value in each case to using the number of live infected cells per ml. For this, the *P_λ_*predicted by the model was multiplied by the number of target cells at the start of infection, which was 10^6^ cells/ml. The best fit for the model as well as the result of the calculation with the experimentally determined *r* and *q* is shown in updated Figures 2 and 3 as Figure 2D (wild type infection) and Figure 3B (L100I mutant infection).

The great improvement of this approach is that *r* values are now constrained: *r* = 0.22 and *q* = 0.17 for wild type HIV cell line infection and *r* = 0.29 and *q* = 0.13 for L100I mutant cell line infection. This now compares well with the experimentally determined *r* = 0.28 and *q* = 0.15. The new analysis is described for wild type infection in the Results section:

“While the percent of infected cells was reduced with drug, the concentration of live infected cells increased (Figure 2—figure supplement 4). […] Equation 3 best fit the behaviour of infection when r=0.22 and q=0.17, resulting in a peak at 4.8nM EFV (Figure 2D, black line).”

It is described for cell line infection with mutant L100I virus in the Results section:

“The fits recapitulated the experimental results when *r* = 0.29 and *q* = 0.13, with a fitted peak at 45nM EFV (Figure 3B, black line). The solution to Equation 3 using the measured values for r and q showed a similar pattern to that obtained with the fitted values (Figure 3B, dashed green line).”

We applied the new model to the coculture lymph node infection results. However, we could not use the number of live infected cells to constrain *r* in this case, since we do not know the number of infectable targets. This is because the lymph node is a complex environment containing, besides CD4^+^ T cells of various kinds and potentially infectable APCs, also CD8^+^ T cells and B cells. This is now discussed in the Results section:

“We did not calculate the predicted number of infected cells for *P_λ_/d* values since the lymph node is a complex environment containing different cell subsets (Sallusto et al., 1999) and the number of infectable target cells at the start of infection is difficult to determine. Hence, we normalized both the experimental number of live infected cells and the *P_λ_/d* values from Equation 3 to the maximum value in each case.”

The results of the new fit are presented in Figure 6B.

For this fit, *r* = 0.91 and *q* = 0.15. While *q* matches the measured cell line result well, the *r* is high. We believe this because it is poorly constrained in the fit, as discussed in the Results section:

“The fits recapitulated the experimental results when *r* = 0.91 and *q* = 0.15, with a fitted peak at 90nM EFV (Figure 6B, black line). […] However, the fitted r value in this case is not expected to be accurate since we were unable to constrain it with the number of infected cells relative to the starting number of target cells.”

2) According to the model q should be independent of the number of attempts. Can this be tested by using different drug concentrations, which would vary the mean number of attempts? If you already have experimental data on this, then include them. Otherwise, just discuss it.

We thank the reviewer for the insightful comments and agree that the probability to die per infection attempt may depend on how many previous infection attempts occurred in the same cell. We discuss this point in the Discussion section:

“Moreover, the probability of death per HIV DNA copy we denote *q* may be dependent on how many infection attempts preceded the current infection attempt, and the model can be improved by measuring this dependence.”

3) In the original manuscript where Equation 1 was derived the authors assumed q was independent of r. In the second paragraph of the Results the authors say q is the probability of a cell to die post reverse transcription, which seems more sensible to me. Under the original assumption of q and r being independent, a cell in which the attempt was unsuccessful would have the same probability of dying as a cell in which reverse transcription succeeded. With q now defined as in the second paragraph of the Results, the fundamental formula given by Equation 1 may not be correct and deserves a more thorough discussion. I think the fundamental process being described is that at each attempt a cell is either infected with probability r, or not infected with probability 1-r. If it is infected, i.e. had a successful reverse transcription, then with probability q it remains alive and with probability 1-q it dies. Assume there are x attempts. Let z be the number of cells infected after x attempts. Then z is binomially distributed, i.e. z= Bin (x,r). Further, if we are interested in the number of cells infected and alive after x attempts, then P(I|x) = Sum_z=1 to x (x choose r) r^z (1-r)^(x-z)(1-q)^z, where the factor (1-q)^z is the probability of a cell surviving after each of the z infections. Simplifying, P(I|x) = [r(1-q) + (1-r)]^x, which is not the same as Equation 1. The number of attempts, x, is again random and as above one can convert this conditional probability into P(I) by assuming a Poisson (or some other distribution such as the empirically measured one) for the number of attempts p(x).

We thank the reviewer for the comment which essentially questions whether the probability of death needs to be independent of the probability to infect for an infection optimum with drug to exist. We have now constructed a second model where death is dependent on infection.

In the case where cell death depends on productive infection, the probability of productive infection and survival of the cell given *n* attempts is shown in Appendix 1-Equation 6.

Here, *r*^0^and *q*^0^are the probabilities for infection and death respectively. Their relationship to the experimentally measured *r* and *q* are discussed in the next section. *r*^0*m*^(1 − *q*^0^)*^m^*is the probability that *m* infection attempts are successful and none of the attempts triggers cell death, while (1 − *r*^0^)*^n^*^−*m*^accounts for *n* − *m* unsuccessful infections.

Realizing that (Appendix 1-Equation 7) we can simplify to Appendix 1-Equation 8.

Averaging this over a Poisson distribution with mean *λ*, we find Appendix 1-Equation 9.

The two models are the same if: (Appendix 1-Equation 10).

In this case the two models are a re-parametrization of the same process. The experimental measurement of *r* and *q*, as described in the main manuscript, consists of estimating *r* as the fraction of infected cells when the fraction of dead cells is low due to a low number of measured DNA copies per cell. *q* is derived by measuring the number of live cells remaining when the number of HIV DNA copies per cell is known. If the probability of death is dependent on infection, then death is only of infected cells and is a product of the probability to be infected and die (*r*^0^*q*^0^). Likewise, the underlying probability of infection *r*^0^will account for the fraction dead cells (*q*) and any cells which are infected but not dead (*r*(1 − *q*)). We therefore conclude that independence of cell death and successful infection is not a required condition for a fitness optimum. This is now stated in the Results section:

“Our model assumes that cellular infection and death due to an HIV infection attempt are independent processes. […] The models are equivalent, showing that independence of cell death and infection is not a necessary condition for an infection optimum to occur in the presence of inhibitor.”

We also discuss the result in the Discussion section:

“Construction of the model assumed independence of productive infection and cell death. However, as shown in the Supplementary Mathematical Analysis section, an equivalent model can be constructed assuming a dependence of cell death on infection.”

The derivation of the above model is included in the new Supplementary Mathematical Analysis section.

4) In Supplementary file 1 the authors say the way they measure the probability a cell is alive after λ attempts is by computing the concentration of cells alive with λ attempts divided by the concentration of cells alive with no transmissions. However, since the concentration of cells alive with no transmissions can approach zero as time goes on it is clear that this fraction need not be less than 1. Also, as noted above the authors need to use conditional probabilities and derive formulas that involve the mean number of attempts. I would suggest they compute the ratio of the probability a cell is infected to the probability a cell in not infected and alive, which is what they measure, and see if they can derive a formula for q.

If we understand the reviewer’s comment correctly, the reviewer’s concern is that the concentration of cells alive with no transmissions approaches zero because both are measured simultaneously in the same culture. However, we set up the experiment differently, where cells equally exposed to all infection conditions except HIV DNA copies were compared for cell death. This was accomplished by treating one of two separate infected cell cultures with EFV at a concentration which resulted in almost complete inhibition of cell-free infection, the mode of infection used. We have now clarified this in the caption of Figure 2—figure supplement 5. Experimental measurement of r and q:

“The concentration of live cells two days post cell-free infection was measured using trypan blue exclusion and divided by the concentration of live reporter cells in a separate cell culture infected with the identical amount of virus, but with infection was inhibited with 40nM EFV.”

We have also included a clarification in Supplementary file 1, which states how the fraction of live cells remaining *L_λ_*is measured.